# Is ImageNet worth 1 video? Learning strong image encoders from 1 long unlabelled video

**Shashanka Venkataramanan**
Inria, Univ Rennes, CNRS, IRISA

**Mamshad Nayeem Rizve**
University of Central Florida

**João Carreira**
Google DeepMind

**Yuki M. Asano***
University of Amsterdam

**Yannis Avrithis***
Institute of Advanced Research
on Artificial Intelligence (IARAI)

## Abstract

Self-supervised learning has unlocked the potential of scaling up pretraining to billions of images, since annotation is unnecessary. But are we making the best use of data? How more economical can we be? In this work, we attempt to answer this question by making two contributions. First, we investigate first-person videos and introduce a "Walking Tours" dataset. These videos are high-resolution, hours-long, captured in a single uninterrupted take, depicting a large number of objects and actions with natural scene transitions. They are unlabeled and uncurated, thus realistic for self-supervision and comparable with human learning.

Second, we introduce a novel self-supervised image pretraining method tailored for learning from continuous videos. Existing methods typically adapt image-based pretraining approaches to incorporate more frames. Instead, we advocate a "tracking to learn to recognize" approach. Our method called DoRA, leads to attention maps that **D**isc**o**ver and t**RA**ck objects over time in an end-to-end manner, using transformer cross-attention. We derive multiple views from the tracks and use them in a classical self-supervised distillation loss. Using our novel approach, a single Walking Tours video remarkably becomes a strong competitor to ImageNet for several image and video downstream tasks. Dataset and code can be found at https://shashankvkt.github.io/dora.

## 1 Introduction

*(To the question "Have you read all the books in here?")*
No, only four of them. But I read those very, very carefully.

Jacques Derrida

Learning from large scale datasets has been at the core of great progress. In particular, the field of self-supervised learning has allowed pretraining of neural networks to scale beyond the size of labelled datasets. By avoiding costly annotation, strong performance has been demonstrated by increasing the training dataset sizes into billions of images.

But how well are those images really used? At a rate of one image per second, a dataset of 1B images would take 317 years to watch. Yet, humans develop functioning visual systems *much faster*.* Besides potential genetic visual priors in humans, one stark difference is the *type* of data. Humans observe their visual surroundings in one continuous stream, only interrupted by sleep. Indeed, learning visual representations of images from videos is not new. However, previous works have found significant gaps in performance to image-pretrained models. They have mostly used object-centric videos scraped from the internet, and adapted image-based pretraining methods to use different frames as an extra form of data augmentation (Gordon et al., 2020; Parthasarathy et al., 2022).

---

*Equal last authors. Order determined randomly

*Humans develop face recognition (de Haan et al., 2001) and color sensitivity (Adams, 1987) in three months, depth perception in five months (Campos et al., 1978) and visual acuity in six months (Sokol, 1978).

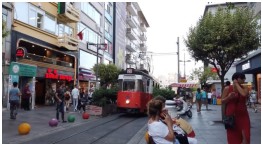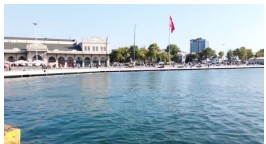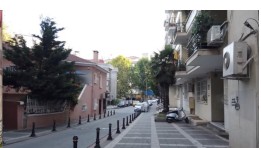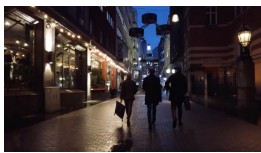

Figure 1: *Examples of frames from the Walking Tours dataset*, containing hours-long, continuous egocentric 4K videos from urban scenes in different cities, under CC-BY license. There are a large number of objects and actions in a variety and natural transition of places, *e.g.* residential area, park, market, waterfront, *etc.*, with natural transition of lighting conditions and object augmentations.

In this work, we investigate two directions. First, in the direction of *data*, we introduce a new dataset of open-source first-person videos, recorded for the purpose of virtual "walking tours", inspired by (Wiles et al., 2022). These videos have several advantages. Not only are the individual frames dense in semantic categories – much more so than movies, as we analyze – but these videos also directly represent the viewpoint of a human, contain few or no shot cuts nor special effects and are long (1-3h). Another benefit is their transparency: indeed, one can watch the whole dataset in one setting. The dataset we create contains 10 Walking Tours (WT) videos with CC-BY license.

Second, in the direction of the *method*, we develop a new self-supervised image-pretraining method that is uniquely suited for learning from natural, non-object-centric videos. Our approach is inspired by observing toddlers first learn to track objects and animals, then to recognize and differentiate them (Bomba & Siqueland, 1983; Quinn et al., 1993; Spelke & Kinzler, 2007). Our method, called DORA, is an end-to-end training approach that "tracks to learn to recognize": given a video clip, objects in an initial frame are implicitly **D**isc**o**vered and t**RA**cked across time. The tracked objects are incentivized to be diverse by introducing a Sinkhorn-Knopp clustering of patch embeddings; the tracked instances are used as a learning signal for a classical multi-view SSL loss.

Surprisingly, contrary to previous works, we find that our novel method obtains ImageNet-level performances by training on *a single WT video*, as evidenced by performances on segmentation and object detection downstream tasks. While humorously intentioned, Derrida's quote rings true to this finding and our results give some hope for alternative directions in SSL that depart from blind dataset scaling towards more efficient and smarter use of existing video data.

To summarize, our key contributions in this work are as follows:

1. We introduce a new dataset of 10 WT videos, with single-video and mixed-video splits. The latter is conveniently equal in size to ImageNet. We analyze their usefulness compared to existing video and image datasets.
2. We propose a new end-to-end self-supervised visual pretraining method called DORA. It builds upon DINO but is tailored to promote tracking of multiple objects across frames. We use it to learn strong image encoders and trace the source of its improvements through extensive ablations.
3. We obtain strong performance on ADE20k segmentation and MS COCO detection, outperforming ImageNet-pretrained DINO, while instead pretraining on a single long video.

## 2 RELATED WORK

Self-supervised learning of image encoders from video data is a very active area of research. Video, and more generally temporal streams, have long been theorized to be ideal signals for unsupervised learning (Wiskott & Sejnowski, 2002). In computer vision, early methods have been very diverse and included pretext tasks such as egomotion prediction (Agrawal et al., 2015; Jayaraman & Grauman, 2015), active recognition (Jayaraman & Grauman, 2016), pose estimation (Chakraborty & Namboodiri, 2017), unsupervised object discovery (Croitoru et al., 2017), dense prediction (Pathak et al., 2017; Li et al., 2019), optical flow (Mahendran et al., 2018; Xiong et al., 2021), frame order prediction (Misra et al., 2016), view-point matching (Sermanet et al., 2018; Pirk et al., 2020) or learning visual correspondences (Wang et al., 2019).

More recently, there have been considerable advances in self-supervised learning using ImageNet, with the main theme being extracting multiple augmentations of an image (Chen et al., 2020; Caron

| DATASET | DOMAIN | EGO | PRE | BAL | ANNOT | AVG. DUR (SEC) | DUR (HR) | #VIDEOS | FRAME RESOLUTION |
|---|---|---|---|---|---|---|---|---|---|
| *Diverse Pretraining* | | | | | | | | | |
| Kinetics-400 (Kay et al., 2017) | Actions | ✗ | ✓ | ✓ | **Class** | 10.2 | 851 | 400 | 340 × 255 |
| WebVid-2M (Bain et al., 2021) | Open | ✗ | ✓ | ✗ | **Weak** | 18 | 13k | – | 320 × 240 |
| HowTo100M (Miech et al., 2019) | Instructions | ✗ | ✓ | ✗ | **Weak** | 4 | 135k | – | – |
| *Egocentric* | | | | | | | | | |
| Epic-Kitchens (Damen et al., 2022) | Cooking | ✓ | ✗ | ✗ | **Loc.** | 510 | 100 | 37 | 1920 × 1080 |
| Ego-4D (Grauman et al., 2022) | Daily | ✓ | ✗ | ✗ | **Loc.** | 1446 | 120 | 931 | 1920 × 1080 |
| Meccano (Ragusa et al., 2023) | Industry | ✓ | ✗ | ✗ | **Loc.** | 1247 | 849 | 20 | 1920 × 1080 |
| Assembly-101 (Sener et al., 2022) | Assembly | ✓ | ✗ | ✗ | **Loc.** | 426 | 167 | 362 | 1920 × 1080 |
| *ImageNet-aligned* | | | | | | | | | |
| R2V2 (Gordon et al., 2020) | ImageNet | ✗ | ✓ | ✓ | **Class** | – | – | – | 467 × 280 |
| VideoNet (Parthasarathy et al., 2022) | ImageNet | ✗ | ✓ | ✓ | **Class** | 10 | 3055 | – | |
| Walking Tours (ours) | Urban | ✓ | ✓ | ✗ | **None** | 5880 | 23 | 10 | 3840 × 2160 |

Table 1: *Walking Tours* vs. *existing video datasets*. EGO: egocentric; PRE: used for pretraining; BAL: class balance control; ANNOT: annotation type. Weak: associated data per clip (text or other modality); Class: class label per frame or clip; Loc: localization per frame (*e.g.* bounding box, segmentation, mask 3D pose). AVG. DUR: average duration per video; DUR: total duration.

et al., 2021) and training models to pull them together/apart. These methods have since percolated to learning from video frames (Gordon et al., 2020; Parthasarathy et al., 2022; Tschannen et al., 2020; Wang & Gupta, 2015; Orhan et al., 2020). Similar to this work, TimeTuning (Salehi et al., 2023) leverages the passage of time in videos by not treating it as simple augmentations. However, in contrast to our work, it requires an already image-pretrained backbone. VITO (Parthasarathy et al., 2022) improves performance relative to ImageNet, by using VideoNet, a large YouTube dataset of 10s videos from a similar class distribution and the same number of examples as ImageNet. In this paper, we show that it is possible to obtain strong results from a *single* long video, with a very different visual distribution compared to ImageNet / VideoNet.

# 3 WALKING TOURS DATASET

## 3.1 DATASET COLLECTION AND PROPERTIES

We collect from YouTube a new dataset of urban scenes called "Walking Tours" (WTours, or WT) comprising 10 egocentric videos of a person walking in different cities in Europe and Asia. The cities include Amsterdam, Bangkok, Chiang Mai, Istanbul, Kuala Lampur, Singapore, Stockholm, Venice, and Zurich. We also include a video from a Wildlife safari. Examples are shown in Figure 1. These videos are captured in 4K resolution (3840 × 2160 pixels) at 60 frames-per-second and are under Creative Commons License (CC-BY). The minimum video duration is 59 minutes (Wildlife safari), the maximum is 2 hours 55 minutes (Bangkok) and the average is 1 hour 38 minutes. Such videos are particularly interesting for visual learning because of the following properties:

1. *Large number of objects and actions*. Each frame or clip taken from a video depicts several objects and actions, *e.g.* walking, riding a bike, sitting, drinking *etc*.
2. *Natural transition in lighting conditions*. In some videos, the lighting gradually transitions from bright (late afternoon) to dim (dusk) then to dark (post sunset).
3. *Natural transition in scenes*. The videos depict transitions between places, *e.g.* from city center to market place to residential areas to parks to water fronts *etc*.
4. *Natural object augmentations*. Continuous variation *e.g.* of pose, deformation, viewpoint, perspective distortion, relative object position, occlusion, background clutter.

The abundance of information within these videos, encompassing a multitude of objects and complex scenes, presents a formidable challenge for manual annotation or curation, making it appropriate for unsupervised pretraining. To the best of our knowledge, we are the first to propose an egocentric video dataset for pretraining and evaluate it on a wealth of downstream tasks.

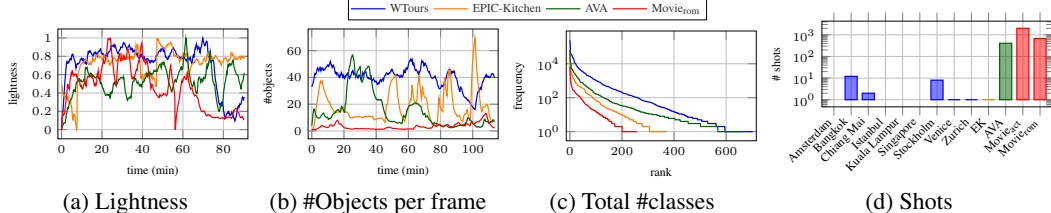

(a) Lightness     (b) #Objects per frame     (c) Total #classes     (d) Shots

Figure 2: *Dataset analysis* of a WTours video compared with videos from Epic-Kitchens (Damen et al., 2022), AVA (Gu et al., 2018) and two entire movies, concatenated or cropped to match the duration of the WTours video. (a) Lightness *vs*. time. (b) Number of objects per frame *vs*. time. (c) Frequency of classes in entire video. (d) Number of shots. Objects detected by Detic (Zhou et al., 2022b), trained on ImageNet-21k.

## 3.2 COMPARISON WITH OTHER VIDEO DATASETS

In Table 1, we compare WTours with existing video datasets. In summary, self-supervised pre-training on videos has been mostly limited to short, low-resolution datasets that rely on weak annotation in the form of video-text pairs (Bain et al., 2021; Miech et al., 2019) or are curated, *e.g.* their class balance is controlled, even if their annotation is unused (Kay et al., 2017). *ImageNet-aligned* datasets (Gordon et al., 2020; Parthasarathy et al., 2022) contain short videos that are (semi-automatically) curated and annotated with the same distribution and classes as ImageNet. *Egocentric* video datasets (Damen et al., 2022; Grauman et al., 2022; Sener et al., 2022) have long, high-quality videos, but are the result of significant manual work. In this paper we aim to learn from videos publicly available online.

WTours videos are continuous, longer and higher-resolution than even other egocentric datasets. Using object detectors, we find that the average number of object classes is close to that of ImageNet and there is a high number of objects per frame, making WTours appropriate for representation learning. WTours is not curated and does not rely on search terms. It is *data-first* and more open-ended, thus well suited for the self-supervised setting. It is scalable since it requires no human labeling effort and more videos can be easily downloaded or even made. We are inspired by a 10k walking tours videos created by Wiles et al. (2022), which however is not publicly released and not studied for self-supervised learning. A more detailed discussion is given in subsection A.1.

## 3.3 DATASET ANALYSIS

In Figure 2, we analyse the properties of a single WTours video compared with videos of the same length from two other datasets, as well as two movie videos. In summary, our findings are as follows. From Figure 2(a), WT may exhibit gradual shifts in lightness, transitioning from bright to dim to dark, while Epic-Kitchens and AVA videos exhibit random brightness fluctuations. Lightness variations are not well expored in self-supervised pretraining. From Figure 2(b,c), unique classes appear more frequently and there are more unique objects per frame in WTours than in the other datasets. This makes WTours semantically richer. From Figure 2(d), WTours and Epic-Kitchens videos contain only one or two shots per entire video on average, while the other datasets contain hundreds. In subsection 5.2 and in Appendix C, we show that WTours significantly outperforms movies in downstream tasks, which is partially attributed to the absence of cuts. More detailed discussion of dataset analysis is given in subsection A.2.

## 4 ATTENTION-BASED MULTI-OBJECT TRACKING

Our goal is to build robust representations by leveraging the rich information in video frames. Standard SSL frameworks (Chen et al., 2020; Caron et al., 2020) often assume correspondences between different views. This is true whether using dense (Zhou et al., 2022a) or global representations by pooling (Caron et al., 2021). While it is relatively straightforward to establish correspondences in images, it becomes more challenging when dealing with temporal deformations, requiring some form of object tracking (Salehi et al., 2023). In videos with a large field of view or ego-motion, obtaining correspondences becomes even more difficult.

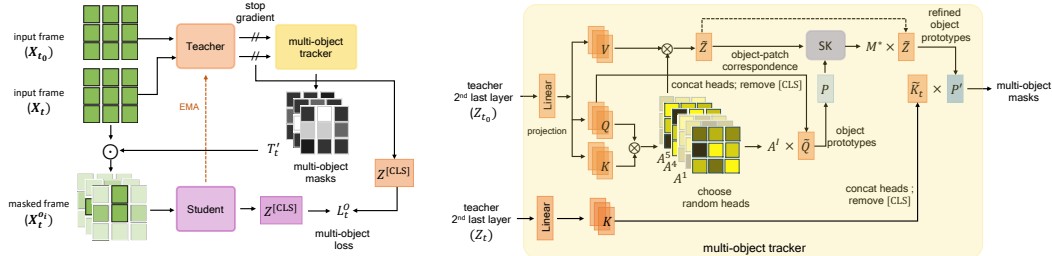

Figure 3: DORA, *our self-supervised image pretraining method from video*. (Left) From an input frame $\mathbf{X}_{t_0}$, the output of the second-last layer of the teacher model is used by a multi-object tracker to generate cross-attention maps $T'_t$ with frame $\mathbf{X}_t$. We use those to mask $\mathbf{X}_t$ (7), feed it to the student model and apply a *distillation loss* $L^{\mathrm{O}}_t$ between [CLS] token embeddings (8). (Right) In the tracker, we obtain the query $Q$, key $K$ and output $Z$ embeddings. From the multi-head attention maps $A^i$ (1), we draw a subset $\mathcal{I}$ of $k$ heads and form object prototypes $P$ by pooling over patch queries $\tilde{Q}$ (2). We refine them into $P'$ to *discover distinct objects*, using Sinkhorn-Knopp (SK) to establish correspondences $M^*$ between $P$ and patch embeddings $\tilde{Z}$ (4) and pooling over $\tilde{Z}$ (5). We then *track the objects* over frames $\mathbf{X}_t$ by cross-attention $T'_t$ with patch key embeddings $\tilde{K}_t$ (6).

**High-level idea**   We introduce DORA, based on multi-object **Di**s**co**very and t**RA**cking. As shown in Figure 3, it leverages the attention from the [CLS] token of distinct heads in a vision transformer to identify and consistently track multiple objects within a given frame across temporal sequences. On these, a teacher-student distillation loss is then applied. Importantly, we do not use any off-the-shelf object tracker or optical flow network. This keeps our pipeline simple and does not require any additional data or training. It also ensures that the learned representation is robust.

**Preliminaries**   We are given a video clip consisting of $T$ frames $\mathbf{X}_t \in \mathbb{R}^{h \times w \times c}$ for $t \in \{1, \ldots, T\}$, where $h \times w$ is the spatial resolution and $c$ is the number of channels. Each frame is split into $n = hw/p^2$ non-overlapping patches of resolution $p \times p$. The patches are linearly projected into embeddings of dimension $d$ and a [CLS] token embedding is prepended. This representation is input to a *transformer encoder* (Dosovitskiy et al., 2020). The *output embeddings* are $Z_t = g_\theta(\mathbf{X}_t) \in \mathbb{R}^{(n+1) \times d}$, where mapping $g_\theta$ includes the tokenizer and encoder, while $\theta$ denotes its learnable parameters. Given an embedding $Z \in \mathbb{R}^{(n+1) \times d}$, we write $Z = [Z^{[\mathrm{CLS}]}; \tilde{Z}]$, where $Z^{[\mathrm{CLS}]} \in \mathbb{R}^{1 \times d}$ is the [CLS] token embedding and $\tilde{Z} \in \mathbb{R}^{n \times d}$ are the *patch embeddings*.

Following DINO (Caron et al., 2021), there is a student network with parameters $\theta$ and a teacher network with identical architecture and parameters $\theta'$ obtained as the *exponential moving average* (EMA) of $\theta$ according to $\theta' \leftarrow \alpha\theta' + (1 - \alpha)\theta$. The encoder is followed by a *head* that includes an MLP and a scaled softmax, such that the output token embeddings can be interpreted as probabilities. We denote by $f_\theta$ the mapping that includes the tokenizer, encoder and head.

**Discovering objects with multi-head attention**   Starting at a first frame $\mathbf{X}_{t_0}$, we obtain the query and key embeddings $Q, K \in \mathbb{R}^{(n+1) \times d}$ from the last transformer layer of the teacher network[†]. According to *multi-head* attention, these embeddings are partitioned as $Q = [Q^1, \ldots, Q^h]$, $K = [K^1, \ldots, K^h]$, where $Q^i, K^i \in \mathbb{R}^{(n+1) \times d/h}$ for $i = 1, \ldots, h$ and $h$ is the number of heads. For each head $i$, the *self-attention* matrix $A^i \in \mathbb{R}^{(n+1) \times (n+1)}$ is based on the dot-product similarity between the query and key embeddings:

$$A^i := \mathrm{softmax}\left(Q^i (K^i)^\top / \sqrt{d}\right) \in \mathbb{R}^{(n+1) \times (n+1)}. \tag{1}$$

Given an attention matrix $A \in \mathbb{R}^{(n+1) \times (n+1)}$, let $A^{[\mathrm{CLS}]} := [a_{1,2}, \ldots, a_{1,n}] \in \mathbb{R}^{1 \times n}$ be the [CLS]-*attention* vector between the [CLS] and patch embeddings, where $a_{i,j}$ is the element $(i, j)$ of $A$. We draw at random a subset $\mathcal{I} := \{i_1, \ldots, i_k\}$ of $k < h$ heads and collect their [CLS]-attention vectors into $A^{\mathcal{I}} := [(A^{i_1})^{[\mathrm{CLS}]}; \ldots; (A^{i_k})^{[\mathrm{CLS}]}] \in \mathbb{R}^{k \times n}$. Intuitively, as expressed in rows of matrix $A^{\mathcal{I}}$, the different heads attend to different *objects* in the frame (Caron et al., 2021).

---

[†]For simplicity, we drop $t_0$ from the notation.

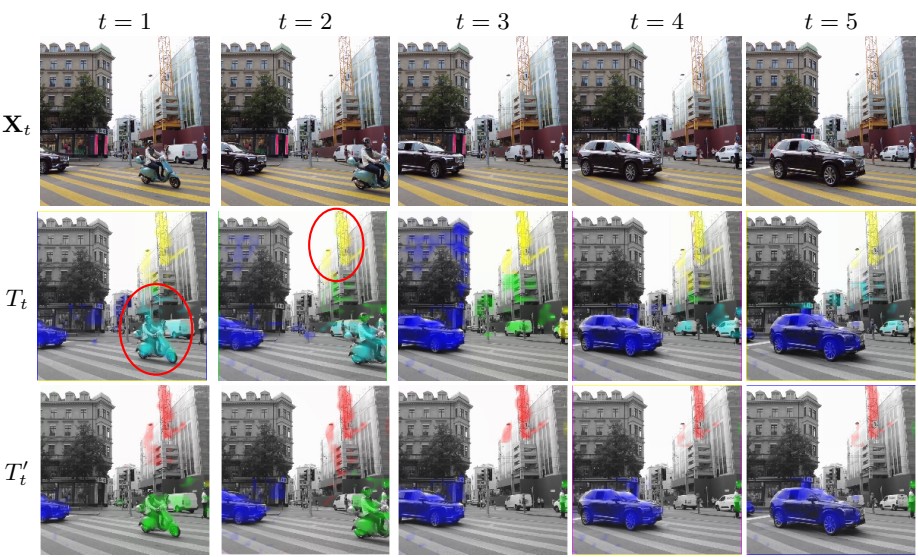

Figure 4: For each input frame $t$ of a video clip (top), cross-attention map $T_t \in \mathbb{R}^{k \times n}$ (3) (middle) and refined cross-attention map $T_t' \in \mathbb{R}^{k \times n}$ (6) (bottom), using Sinkhorn-Knopp algorithm. For each object, one row of $T_t$ or $T_t'$ is reshaped as $h/p \times w/p$ and upsampled to an $h \times w$ *attention map* overlaid on the input frame for $k = 3$ objects encoded in blue, red and green channel. Mixed colors yellow and cyan for $T_t$ (middle, in red circle) indicate spatial overlap of two objects, while $T_t'$ (bottom) yields three well separated objects shown in primary colors blue, red and green.

To represent the $k$ objects in the embedding space, we use matrix $A^{\mathcal{I}} \in \mathbb{R}^{k \times n}$ to form linear combinations of patch embeddings $\tilde{Q} \in \mathbb{R}^{n \times d}$, obtaining *object prototypes*

$$P := A^{\mathcal{I}} \tilde{Q} \in \mathbb{R}^{k \times d}. \tag{2}$$

This can be seen as the representation of $k$ different [CLS] tokens in the full embedding space, capturing $k$ objects at frame $t_0$. Then, given the key embeddings $K_t \in \mathbb{R}^{(n+1) \times d}$ at another frame $t$, we could track the objects by *cross-attention*

$$T_t := \mathrm{softmax}\left( P \tilde{K}_t^\top / \sqrt{d} \right) \in \mathbb{R}^{k \times n}, \tag{3}$$

where $\tilde{K}_t \in \mathbb{R}^{n \times d}$. Unfortunately, we observe in Figure 4 that the $k$ attention maps obtained this way are spatially overlapping, meaning that each attention map is not delineating a single object.

**Establishing object-patch correspondences**    To discover spatially distinct objects, we propose to establish correspondences between prototypes and patch tokens. Let $Z = g_{\theta'}(\mathbf{X}_{t_0}) \in \mathbb{R}^{(n+1) \times d}$ be the output embeddings of the teacher network, still at frame $t_0$. We seek a correspondence between the rows of $P \in \mathbb{R}^{k \times d}$ and $\tilde{Z} \in \mathbb{R}^{n \times d}$, where $\tilde{Z}$ are the patch token embeddings.

The goal is to find a *transport plan* $M \in \mathbb{R}^{k \times n}$ that minimizes the expected pairwise cost $C := -P\tilde{Z}^\top \in \mathbb{R}^{k \times n}$ between prototypes and patches, while incorporating an entropic regularizer with coefficient $\epsilon$. Matrix $M$ is non-negative with row-wise sum $1/k$ and column-wise sum $1/n$, representing a joint probability over $P$ and $\tilde{Z}$ with uniform marginals. The minimal solution $M^*$ is unique and can be found by forming the matrix $e^{-C/\epsilon}$ and then applying the Sinkhorn-Knopp (SK) algorithm (Cuturi, 2013), *i.e.*, iteratively normalizing its rows and columns:

$$M^* = \mathrm{SK}\left( \exp\left( P\tilde{Z}^\top / \epsilon \right) \right) \in \mathbb{R}^{k \times n}, \tag{4}$$

Observe the similarity with (1) and (3), where scaling is by $\sqrt{d}$ rather than $\epsilon$, $\exp$ is included in softmax and normalization is on rows only rather than iterative. Then, similarly with (2), we use the optimal transport plan $M^* \in \mathbb{R}^{k \times n}$ to form linear combinations of patch embeddings $\tilde{Z} \in \mathbb{R}^{n \times d}$, obtaining the *refined object prototypes*

$$P' = M^* \tilde{Z} \in \mathbb{R}^{k \times d}. \tag{5}$$

Now, given the key embeddings $K_t \in \mathbb{R}^{(n+1) \times d}$ at another frame $t$, we track the objects by the *refined cross-attention*, similarly with (3):

$$T_t' := \text{softmax}\left( P' \tilde{K}_t^\top / \sqrt{d} \right) \in \mathbb{R}^{k \times n}, \tag{6}$$

where $\tilde{K}_t \in \mathbb{R}^{n \times d}$. Indeed, Figure 4 confirms that each of the $k$ resulting attention maps is associated with a spatially distinct object, thanks to the established correspondences.

In contrast to previous works that use SK in the context of self-supervised learning to force an equipartitioning of images to cluster labels (Asano et al., 2020; Caron et al., 2020; Oquab et al., 2023), we rather use optimal transport to re-balance *spatial* correspondences to different objects.

**Multi-object masking**    We use the cross-attention (6) to mask the input video clip for the student network, such that each masked clip can be considered as a *multi-object crop*. This crop plays a similar role with local crops in DINO (Caron et al., 2021), but it has arbitrary shape and tracks an object over video frames. In particular, given an input frame $\mathbf{X} \in \mathbb{R}^{h \times w \times c}$ with cross-attention matrix $T' \in \mathbb{R}^{k \times n}$ (6) and an object $i \in \{1, \ldots, k\}$, we reshape the $i$-th row of $T'$ as $h/p \times w/p$ and upsample to a $h \times w$ attention map to match the spatial resolution of $\mathbf{X}$, as shown in Figure 4. We repeat along the channel dimension to form tensor $\mathbf{T}^i \in \mathbb{R}^{h \times w \times c}$ and we mask $\mathbf{X}$ as

$$\mathbf{X}^{o_i} := \mathbf{X} \odot \mathbf{T}^i, \tag{7}$$

where $\odot$ is the Hadamard product. Following DINO (Caron et al., 2021), given an input frame $\mathbf{X}_t$, we generate two standard resolution augmented *global views* $\mathbf{X}_t^a, \mathbf{X}_t^b$. We introduce a *multi-object loss* $L_t^O$ for frame $t$, applied to the [CLS] token between the teacher $f_{\theta'}$ output for one global view $\mathbf{X}_t^u$ and the student $f_\theta$ output for the masked version $\mathbf{X}_t^{v,o_i}$ of the other view $\mathbf{X}_t^v$ for $i \in \{1, \ldots, k\}$, where $u, v \in V = \{a, b\}$ and $u \neq v$:

$$L_t^O := \sum_{u,v \in V} \mathbb{1}_{u \neq v} \sum_{i=1}^k f_{\theta'}(\mathbf{X}_t^u)^{[\text{CLS}]} \log\left( f_\theta(\mathbf{X}_t^{v,o_i})^{[\text{CLS}]} \right). \tag{8}$$

In addition, as detailed in subsection B.1, we apply a *local loss*, following *multi-crop* (Caron et al., 2020). The overall loss $L$ is the sum of the two losses, averaged over all $T$ frames.

## 5 EXPERIMENTS

### 5.1 SETUP

**Tasks and methods**    We perform self-supervised pretraining on a single WT tour video in Venice (referred to as $\text{WT}_{\text{Venice}}$) or all 10 WT videos (referred to as $\text{WT}_{\text{all}}$) and compare with other image and video datasets. To evaluate the quality of the learned representations, we use frozen features for classification, unsupervised object discovery and video object segmentation. We fine-tune for semantic segmentation, object detection and object tracking. We compare DORA with SoTA SSL methods (da Costa et al., 2022) using our settings. We provide more details in individual sections per task. Implementation details and hyperparameters are given in Appendix B.

### 5.2 ABLATIONS

We examine the effect of using different pretraining video dataset and different options and parameters for DORA, measuring performance of classification on ImageNet-1k (Deng et al., 2009) by linear probing (LP) accuracy and unsupervised object discovery on Pascal-VOC 2012 (Everingham et al.) by correct localization (CorLoc) (Siméoni et al., 2021).

**Pretraining video dataset**    We study the impact of pretraining on diverse video datasets, encompassing *object-centric* videos such as Kinetics-400 (K-400) (Kay et al., 2017), *egocentric* videos like Epic-Kitchens (EK) (Damen et al., 2022) and a single movie, Movie$_{\text{rom}}$ (Central). To maintain uniformity in terms of the number of frames, we curate a subset of videos from K-400 and EK, such that their total duration is the same as a single WT video. In Table 2a, we observe that although K-400 is *object-centric*, pretraining on WTours videos yields superior performance on ImageNet and Pascal-VOC 2012. Pretraining on a single movie yields is inferior to both WTours and K-400 by a large margin. This is possibly due to the presence of cuts, as studied in Appendix C.

| METHOD | PRETRAIN | #FRAMES (M) | LP | CORLOC |
|---|---|---|---|---|
| DINO | Movie$_{rom}$ | 0.19 | 34.9 | 51.5 |
| DORA | Movie$_{rom}$ | 0.19 | **35.3** | **51.6** |
| DINO | K-400* | 0.2 | 40.7 | 52.4 |
| DORA | K-400* | 0.2 | **43.0** | **55.2** |
| DINO | EK* | 0.2 | 38.6 | 53.5 |
| DORA | EK* | 0.2 | **41.8** | **56.0** |
| DINO | WT$_{Venice}$ | 0.2 | 33.8 | 51.2 |
| DORA | WT$_{Venice}$ | 0.2 | **44.5** | **56.2** |

(a) *Video datasets*

| METHOD | $k$ | LP | CORLOC |
|---|---|---|---|
| DINO | ✗ | 33.8 | 51.2 |
| DORA | 1 | 39.9 | 53.9 |
| DORA | 2 | 43.1 | 55.7 |
| DORA | 3 | **44.5** | **56.2** |
| DORA | 4 | 39.2 | 53.8 |
| DORA | 5 | 36.7 | 50.3 |
| DORA | 6 | 35.8 | 48.8 |
| DORA | 16 | 28.3 | 48.5 |
| DORA | 32 | 27.1 | 46.8 |

(b) *#Objects $k$ on WT$_{Venice}$*

| METHOD | SK | MASK | LP | CORLOC |
|---|---|---|---|---|
| DINO | ✗ | ✗ | 33.8 | 51.2 |
| DORA | ✗ | Random | 33.0 | 49.8 |
| DORA | ✗ | Object | 42.5 | 55.3 |
| DORA | ✓ | Random | 29.9 | 46.7 |
| DORA | ✓ | Object | **44.5** | **56.2** |

(c) *SK and masking on WT$_{Venice}$*

Table 2: *Effect of parameters*. ViT-S/16 pretrained, then frozen. (a) Different pretraining video dataset, (b) Number $k$ of tracked objects. (c) Random or multi-object mask, without SK (3) and with SK (6). *: subset of videos with same total duration as a single WTours video. K-400: Kinetics-400, EK: Epic-Kitchens. LP: top-1 accuracy (%) of linear probing on the validation set of ImageNet-1k. CorLoc: correct localization on validation set of Pascal-VOC 2012.

| METHOD | EPOCHS | PRETRAIN | (a) SEMANTIC SEG. | | | | (b) OBJECT DET. | | (c) INSTANCE SEG. | |
|---|---|---|---|---|---|---|---|---|---|---|
| | | | mIoU | GAIN | Acc$_m$ | GAIN | mAP | GAIN | mIoU | GAIN |
| ViT-S/16 | 100 | none | 25.1 | | 33.3 | | 28.6 | | 24.3 | |
| iBOT (Zhou et al., 2022a) | 100 | WT$_{Venice}$ | 33.9 | | 43.3 | | 37.6 | | 33.0 | |
| AttMask (Kakogeorgiou et al., 2022) | 100 | WT$_{Venice}$ | 33.6 | | 42.7 | | 36.5 | | 32.5 | |
| VITO (Parthasarathy et al., 2022) | 300 | VideoNet | 39.4 | | – | | 44.0 | | – | |
| DINO (Caron et al., 2021) | 100 | IN-1k | 33.9 | | 44.3 | | 39.9 | | 35.1 | |
| DORA (ours) | 100 | WT$_{all}$ | 36.9 | | 48.0 | | 40.7 | | 36.3 | |
| DINO (Caron et al., 2021) | 100 | WT$_{Venice}$ | 32.4 | | 43.7 | | 37.1 | | 32.1 | |
| DORA (ours) | 100 | WT$_{Venice}$ | **35.4** | +3.0 | **45.5** | +1.8 | **39.5** | +2.4 | **34.7** | +2.6 |

Table 3: *Semantic segmentation, object detection and instance segmentation*. ViT-S/16 pretrained, then fine-tuned. WT$_{Venice}$: Walking Tours (ours), single video of *Venice*; WT$_{all}$: all videos. IN-1k: ImageNet-1k. (a) Semantic segmentation: fine-tuning on ADE20k using UperNet. mIoU: mean IoU; Acc$_m$: mean-class accuracy. (b) Object detection and (c) Instance segmentation: fine-tuning on MS-COCO using Cascade RCNN. mAP: mean average precision; mIoU: mean IoU.

**Number of tracked objects** We study the impact of the number $k$ of objects. Objects are discovered using attention heads, where the total number of heads is in ViT-S/16 is $h = 6$. For $k > h$, we modify the MSA block as described in subsection B.2. In Table 2b, we observe that $k = 3$ works best. We hypothesize that this is a compromise between the number of objects that can be tracked and the multi-object loss (8) attempting to match small objects with the global crop.

**Choice of masking and Sinkhorn-Knopp** We explore the effect of using a multi-object mask (7) *vs.* random block-wise (Zhou et al., 2022a) and the effect of improving object-patch correspondence through SK in refined cross-attention (6) *vs.* (3). In Table 2c, we observe that a multi-object mask leads to a remarkable performance improvement even in the absence of SK. In fact, random block-wise mask undermines object-patch correspondence, making the effect of SK negative. By contrast, SK improves performance in the presence of multi-object mask.

## 5.3 COMPARISON WITH STATE-OF-THE-ART

**Dense scene understanding** Table 3(a) shows *semantic segmentation* by fine-tuning on ADE20k (Zhou et al., 2017) using UperNet (Xiao et al., 2018). DORA outperforms DINO by 3% mIoU, and 1.8% Acc$_m$. It is interesting to note that DORA pretrained on 200k frames of a *single* WTours video outperforms DINO pretrained on 1.3M images of ImageNet-1k by 1.5% mIoU. A more comparable setting is DORA pretrained on 1.5M frames of WT$_{all}$, which outperforms DINO pretrained on ImageNet by 3% mIoU. Table 3(b) shows *object detection* and *instance segmentation* by fine-tuning on MS-COCO (Lin et al., 2014) using Cascade RCNN (Cai & Vasconcelos, 2019). DORA outperforms DINO by 2.4% mAP and 2.6% mIoU. DORA pretrained on WT$_{all}$ outperforms DINO pretrained on ImageNet by 0.8% mIoU and 1.2% mAP. This shows that pretraining on WTours videos significantly improves the generality of DORA to dense prediction tasks, requiring only one tenth of the total images.

| METHOD | EPOCHS | PRETRAIN | (a) VIDEO OBJECT SEGMENTATION | | | | | | (b) OBJECT TRACKING | | | | | |
|---|---|---|---|---|---|---|---|---|---|---|---|---|---|---|
| | | | $(\mathcal{J}\&\mathcal{F})_m$ | GAIN | $\mathcal{J}_m$ | GAIN | $\mathcal{F}_m$ | GAIN | mAO | GAIN | $SR_{0.5}$ | GAIN | $SR_{0.75}$ | GAIN |
| ViT (Dosovitskiy et al., 2020) | 100 | None | 26.9 | | 25.4 | | 28.3 | | 23.1 | | 19.0 | | 3.4 | |
| iBOT (Zhou et al., 2022a) | 100 | $WT_{Venice}$ | 57.4 | | 56.7 | | 58.0 | | 41.5 | | 47.5 | | 16.6 | |
| DINO (Caron et al., 2021) | 100 | IN-1k | 59.4 | | 57.4 | | 61.4 | | 46.4 | | 54.3 | | 24.1 | |
| DoRA (ours) | 100 | $WT_{all}$ | 57.6 | | 55.1 | | 60.2 | | 45.9 | | 53.4 | | 23.7 | |
| DINO (Caron et al., 2021) | 100 | $WT_{Venice}$ | 54.6 | | 53.0 | | 56.2 | | 37.4 | | 41.4 | | 13.4 | |
| DoRA (ours) | 100 | $WT_{Venice}$ | **58.4** | +3.8 | **56.4** | +3.4 | **60.4** | +4.2 | **41.4** | +4.0 | 47.2 | +5.8 | 18.2 | +4.8 |

Table 4: *Video object segmentation and object tracking*. ViT-S/16 pretrained, then frozen or fine-tuned. $WT_{Venice}$: Walking Tours (ours), single video from *Venice*; $WT_{all}$: all videos. IN-1k: ImageNet-1k. (a) Video object segmentation: frozen features on DAVIS-2017. $\mathcal{J}_m$: mean region similarity; $\mathcal{F}_m$: mean contour-based accuracy. (b) Multi-object tracking: fine-tuning on on GOT-10k. mAO: mean average overlap; SR: success rate, threshold 50% and 75%.

| METHOD | EPOCHS | PRETRAIN | #FRAMES (M) | (a) CLASSIFICATION | | | | (b) OBJECT DISCOVERY | | | |
|---|---|---|---|---|---|---|---|---|---|---|---|
| | | | | LP | GAIN | $k$-NN | GAIN | JACC. | GAIN | CORLOC | GAIN |
| SimCLR (Chen et al., 2020) | 100 | $WT_{Venice}$ | 0.2 | 26.3 | | 25.9 | | 40.4 | | 50.2 | |
| SwAV (Caron et al., 2020) | 100 | $WT_{Venice}$ | 0.2 | 28.0 | | 26.4 | | 40.6 | | 51.4 | |
| iBOT (Zhou et al., 2022a) | 100 | $WT_{Venice}$ | 0.2 | 36.8 | | 32.8 | | 43.0 | | 53.1 | |
| AttMask (Kakogeorgiou et al., 2022) | 100 | $WT_{Venice}$ | 0.2 | 35.8 | | 31.9 | | 43.5 | | 54.5 | |
| VicReg (Bardes et al., 2021) | 100 | $WT_{Venice}$ | 0.2 | 36.5 | | 30.1 | | 42.7 | | 52.1 | |
| DINO (Caron et al., 2021) | 100 | $WT_{Venice}$ | 0.2 | 33.8 | | 29.9 | | 43.8 | | 51.2 | |
| DoRA (ours) | 100 | $WT_{Venice}$ | 0.2 | **45.4** | +11.6 | **33.8** | +3.9 | **44.0** | +0.2 | **56.2** | +5.0 |
| DINO (Caron et al., 2021) | 100 | $WT_{all}$ | 1.5 | 36.6 | | 31.1 | | 42.9 | | 55.8 | |
| DoRA (ours) | 100 | $WT_{all}$ | 1.5 | **45.3** | +8.7 | **35.7** | +4.6 | **44.3** | +1.4 | **57.1** | +1.3 |

Table 5: *Image classification and object discovery*. ViT-S/16 pretrained, then frozen. $WT_{Venice}$: Walking Tours (ours), single video from *Venice*; $WT_{all}$: all videos. (a) Classification top-1 accuracy (%) on validation set of ImageNet-1k. LP: linear probing. (b) Unsupervised object discovery on validation set of Pascal-VOC 2012. Jacc.: Jaccard similarity; CorLoc: Correct Localization.

**Video understanding**  Table 4(a) shows video object segmentation by using frozen features on DAVIS-2017 (Pont-Tuset et al., 2017), which assesses the ability to segment an object over its dynamic temporal changes. DoRA captures detailed temporal deformations and outperforms baseline DINO by 3.4% $\mathcal{J}_m$ and 4.2% $\mathcal{F}_m$. Using only a *single* video for pretraining, DoRA achieves almost the same performance of DINO pretrained on ImageNet (56.4% *vs.* 57.4% $\mathcal{J}_m$). Table 4(b) shows multi-object tracking by fine-tuning on GOT-10k (Huang et al., 2021) using SeqTrack (Chen et al., 2023). GOT-10k assesses the ability to track extremely fast moving objects, objects with illumination variation and low resolution. DoRA achieves significant gains between 4-6% over DINO.

**Image classification and unsupervised object discovery**  We pretrain DoRA on WTours and then we keep it frozen on the downstream task, indicating the quality of the pretrained features. Table 5(a) shows *image classification* on ImageNet-1k, measuring accuracy for linear probing and $k$-nearest neighbor. Table 5(b) shows *unsupervised object discovery* on Pascal-VOC 2012, using attention maps as segmentation masks to measure Jaccard similarity and CorLoc.

On both tasks, non-contrastive methods (DINO, iBOT, VICReg) outperform contrastive methods (SimCLR, SwAV), when pretrained on a single WT video. Importantly, non-contrastive methods are also more efficient to train, since no negative pairs are used. Also on both tasks, DoRA outperforms DINO by a large margin, *e.g.* 11.6% LP and 3.9% $k$-NN on classification, when trained on a single WT video. Comparing DoRA on $WT_{Venice}$ with the $WT_{all}$ dataset, the improvement brought by the full dataset is small when using DoRA, although it is 10 times larger.

In Appendix D, we show that DoRA outperforms SoTA methods on all tasks on ImageNet-1k.

## 6   CONCLUSIONS

We have introduced a dataset of 10 walking tour videos – first-person videos taken by people touring a city, with no cuts, high resolution and that are hours long. We show that learning from clips taken from these videos is surprisingly powerful: with an appropriately tailored self-supervised learning method for videos, we obtain representations that rival those obtained on ImageNet when

transferring to popular downstream image and video tasks. This differs from previous state-of-the-art approaches to learning image encoders from video, which also obtain such results but require large video datasets, following closely the ImageNet blueprint.

Our proposed learning method DORA is inspired by DINO, generalizing it to video by incorporating implicit multi-object tracking across video clips. We observe that the method leads to interesting emergent attention masks within the transformer model, that seem to latch on to particular objects, even through occlusions. This makes it uniquely suited to our newly introduced dataset.

## 7 ACKNOWLEDGEMENTS

This work was in part supported by the ANR-19-CE23- 0028 MEERQAT project and was performed using the HPC resources from GENCI-IDRIS Grant 2021 AD011012528. We also thank Dilara Gokay and Andrew Zisserman for their valuable feedback on this paper.

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

# Is ImageNet worth 1 video? Learning strong image encoders from 1 long unlabelled video

## Supplementary Material

## Table of Contents

## A  MORE ON WALKING TOURS

### A.1  COMPARISON WITH OTHER VIDEO DATASETS

In Table 1, we compare WTours with different types of existing video datasets. Self-supervised pretraining on videos has been mostly limited to video datasets that rely on weak annotation in the form of video-text pairs (Bain et al., 2021; Miech et al., 2019) or even are curated, *e.g.* their class balance is controlled, even if their annotation is unused (Kay et al., 2017). Their average clip duration is small, *e.g.* less than 20 sec, and their resolution is also small, limiting the capacity to detect objects at a greater distance. By contrast, WTours videos are continuous, hours-long at high resolution and provide natural transitions of scenes and viewing conditions. They are not curated and thus better suited for the self-supervised setting.

*ImageNet-aligned* datasets such as R2V2 (Gordon et al., 2020) and VideoNet (Parthasarathy et al., 2022) contain videos that are curated and annotated with the same distribution and classes as ImageNet, meant for pretraining image encoders. These videos are short, *i.e.* 10 seconds on average. By contrast, WTours consists of a continuous stream of egocentric video, where the average number of classes is close to that of ImageNet, as shown in subsection 3.3. The rich information contained in 4K resolution, together with a high number of objects in a frame, makes it appropriate for representation learning. Importantly, the continuity and absence of curation make it more realistic and more comparable with human learning. Our dataset does not rely on a set of objects, human activities or other search terms but instead is *data-first* and more open-ended.

Despite the large number of high-quality videos, *egocentric* video datasets (Damen et al., 2022; Grauman et al., 2022; Sener et al., 2022) have been used only for downstream tasks and thus come with extensive annotation. In comparison, WTours has 4-10 times longer average duration and twice the frame resolution. While WTours is smaller in terms of total duration and number of videos, it is scalable under the self-supervised setting since it requires no human labeling effort and more videos

can be easily found, downloaded or even made. This makes collecting more data as simple as a walk in the park.

*Very long video datasets.* A large dataset of 10k WTours videos was created recently by (Wiles et al., 2022) but was not publicly released and not studied for self-supervised learning. Another dataset having hour-long videos is introduced in (Khan et al., 2020), in the context of sports analytics; it has not been explored for self-supervised learning either.

### A.2 DATASET ANALYSIS

Here we present a more detailed discussion on the dataset analysis of subsection 3.3. We refer to Figure 2, where we analyse the properties of a single WTours video compared with videos of the same length from Epic-Kitchens (Damen et al., 2022) and AVA (Gu et al., 2018) datasets, as well as two movie videos, an action movie and a romantic movie.

**Variation in lightness** We measure the change in perceived brightness using the lightness value (L) across consecutive frames. From Figure 2(a), we observe a gradual shift at roughly 150 min into the WTours video, transitioning from bright to dim to dark. By contrast, Epic-Kitchens and AVA videos exhibit random brightness fluctuations, alternating between dim and bright conditions. Typically, self-supervised pretraining happens on datasets with uniform brightness levels. Datasets featuring such brightness variations are less expored.

**Variation in number of objects** Using Detic (Zhou et al., 2022b), a DETR-style object detector trained on ImageNet-21k, we detect objects in each frame. Figure 2(b) shows the number of unique objects per frame and Figure 2(c) shows their frequency in the entire video. We observe that WTours contains 703 unique objects, while Epic-Kitchens has 373, AVA has 663 and Movie-2 has 259. The unique objects appear more frequently and there are more unique objects per frame in WTours than in the other datasets. This makes WTours semantically richer, despite coming from one continuous stream of video. Using videos with a large number of objects can encourage the model to capture complex relations and variations in the data.

**Variation in shots** Egocentric videos are typically captured in a single uninterrupted take, with exceptions being post-processed special effects or cuts. In Figure 2(d), we find that, on average, WTours and Epic-Kitchens videos contain only one or two shots per entire video, while AVA contains 406, an action movie (Movie$_{\text{act}}$) (Skiptrace) contains 2000 and a romantic movie (Movie$_{\text{rom}}$) (Central) contains 667. The substantial number of shots in movies and AVA poses challenges for representation learning methods that rely on object tracking or optical flow. In subsection 5.2, we show that WTours significantly outperforms movies in downstream tasks, which may be attributed to the absence of cuts.

## B MORE ON EXPERIMENTAL SETUP

### B.1 MULTI-CROP

Following DINO (Caron et al., 2021) and iBOT (Zhou et al., 2022a), we apply the *multi-crop* strategy (Caron et al., 2020). In particular, we generate $m$ *local crops* $\mathbf{X}_t^{\ell_i}$ of smaller resolution for $i \in \{1, \ldots, m\}$. The *local loss* $L_t^{\text{LC}}$ for frame $t$ is applied to the [CLS] token between the teacher $f_{\theta'}$ output for a global view $\mathbf{X}_t^u$ and the student $f_\theta$ output for the local crop $\mathbf{X}_t^{\ell_i}$ for $i \in \{1, \ldots, m\}$:

$$L_t^{\text{LC}} := \sum_{v \in V} \sum_{i=1}^{m} f_{\theta'}(\mathbf{X}_t^v)^{[\text{CLS}]} \log\left(f_\theta(\mathbf{X}_t^{\ell_i})^{[\text{CLS}]}\right) \tag{9}$$

The overall loss $L$ is the sum of the multi-object loss $L_t^{\text{O}}$ (8) and the local loss $L_t^{\text{LC}}$ (9), averaged over all $T$ frames:

$$L := \frac{1}{T} \sum_{t=1}^{T} (L_t^{\text{O}} + L_t^{\text{LC}}). \tag{10}$$

## B.2 Implementation details

We use ViT-S/16 (Dosovitskiy et al., 2020) as the backbone in all our experiments. For each mini-batch, we randomly sample clips from the video, consisting of $T = 8$ frames temporally separated by 1 second *i.e.* we sample one frame every 30. Objects discovered in the first frame are tracked over the following 7 frames. Since each frame contains several different objects, applying the standard multi-crop augmentation (Caron et al., 2020) to the entire frame would result in crops with very different visual content or *noisy* positive pairs. Instead, we apply multi-crop to a $300 \times 300$ crop that we first take from the frame. Following DINO (Caron et al., 2021), we obtain two *global crops* and six *local crops*. Masking (7) is applied to the global crops seen by the student for the multi-object loss (8), while local crops are seen directly by the student for the local loss (9). We train for 100 epochs by default.

Objects are discovered using attention heads, where the total number of heads is in ViT-S/16 is limited to $h = 6$. For the purpose of the ablation of the number $k$ of objects for $k > h$ in Table 2b, we modify the MSA block in the final layer, resulting in configurations of 16 and 32 heads. Consequently, we can identify and track up to 16 and 32 objects within the video clip. To accomplish this, we decompose the query and key embeddings of dimension $d = 768$ into 16 and 32 subvectors, resulting in new feature dimensions of 24 and 12 respectively, as opposed to 64 for 6 heads. In Table 2b, we observe that tracking 16 or 32 objects results in overall poor performance possibly due to the small feature dimension, which encodes poor representations.

## B.3 Hyperparameters

**ImageNet-1k: Linear probing and $k$-NN** We pretrain DORA in a self-supervised setting with ViT-S/16 using DINO for 100 and 300 epochs. We use two global and six local crops for each clip and train on 8 A100 GPUs with a global batch size of $16 \times 8 = 128$. We use LARS with a learning rate of $5 \times 10^{-4}$, minimum learning rate of $1 \times 10^{-6}$, global crop scale of $[0.4, 1.0]$ and local crop scale $[0.05, 0.4]$.

For linear probing, we follow (Caron et al., 2021) and use the frozen features of the transformer backbone to train a linear classifier in a supervised setting. We use global batch size of 1024 on the training set and evaluate on the validation set of ImageNet-1k. We use top-1 accuracy (%) as our evaluation metric. For $k$-NN, we freeze the backbone and extract features of training images, then use a $k$-nearest neighbour classifier with $k = 20$.

**Pascal-VOC 2012: Object discovery** We use the validation set of Pascal VOC 2012 (Everingham et al.), which comprises a total of 1449 images. Following LOST (Siméoni et al., 2021), we use the averaged self-attention map, extracted from the final layer of a our pretrained ViT-S/16, to retain $80\%$ of the mass. We use the Jaccard similarity $J$ measured as overlap between predicted mask $P$ and the ground truth mask $G$ as $J(P, G) = \frac{G \cap P}{G \cup P}$. We also use CorLoc, which measures the number of correct predicted boxes, where a predicted box is said to be correct if its IoU $\geq 0.5$.

**ADE20k: Semantic segmentation** We evaluate DORA on ADE20k (Zhou et al., 2017) for semantic segmentation. The dataset includes 20,000 images in the training set and 2,000 images in the validation set. We use UperNet (Xiao et al., 2018) as the segmentation model and use DORA pretrained on WT to initialize the backbone. Following the experimental settings in iBOT (Zhou et al., 2022a), we use AdamW (Loshchilov & Hutter, 2019) with an initial learning rate of $6 \times 10^{-5}$, weight decay of $1 \times 10^{-2}$, and linear warmup of 1,500 iterations. We fine-tune for 160,000 iterations with a batch size of 4.

**MS-COCO: Object detection** We evaluate DORA for object detection and instance segmentation on MS-COCO. We use Cascade Mask R-CNN (Cai & Vasconcelos, 2019), which produces bounding boxes and instance masks simultaneously on the COCO dataset. We use a multi-scale training strategy, where we resize images to have a shorter side ranging between 480 and 800, ensuring that the longer side does not exceed 1,333 pixels. The learning rate is $1 \times 10^{-4}$ and the weight decay is 0.05. During training, we fine-tune the entire network using a $1\times$ schedule, which involves 12 epochs with learning rate reductions by a factor of 10 at epochs 9 and 11. We explore different layer decay rates, specifically $0.65, 0.75, 0.8, 0.9$, with a rate of 1.0 indicating no decay.

| VIDEO | LP | CORLOC |
|---|---|---|
| Amsterdam | **45.4** | 54.5 |
| Bangkok | 42.1 | 54.3 |
| Chiang Mai | 44.9 | 55.5 |
| Istanbul | 44.5 | 54.6 |
| Kuala Lampur | 43.9 | 54.1 |
| Singapore | 42.7 | 54.7 |
| Stockholm | 44.1 | 54.7 |
| Venice | 44.5 | **56.2** |
| Wildlife | 44.0 | 54.9 |
| Zurich | 44.9 | 54.4 |
| Mean | 44.1 | 54.8 |

(a) *WT videos*

| METHOD | PT | LP | CORLOC |
|---|---|---|---|
| DINO | WT | 33.8 | 51.2 |
| DORA | Movie | 35.3 | 51.6 |
| DORA | Movie[†] | 39.8 | 54.8 |
| DORA | WT | **44.5** | **56.2** |

(b) *Cuts*

Table 6: *Effect of pretraining video and cuts*. ViT-S/16 pretrained, then frozen. (a) Different WTours video, using DORA. (b) Effect of cuts. [*]: subset of videos with same total duration as a single WTours video. [†]: sampling without cuts. LP: top-1 accuracy (%) of linear probing on the validation set of ImageNet-1k. CorLoc: correct localization on validation set of Pascal-VOC 2012.

To generate hierarchical feature maps, we utilize the features produced by layers 4, 6, 8, and 12 of our network and apply two deconvolutions for layer 4, one deconvolution for layer 6, identity mapping for layer 8, and max-pooling for layer 12. These post-processing steps enable the creation of hierarchical feature representations. It is important to note that we do not employ multi-scale testing in our experiments.

**DAVIS-2017: Video object segmentation**  We assess the performance of DORA for video object segmentation on DAVIS 2017 dataset (Pont-Tuset et al., 2017), which involves segmenting between 2 to 4 objects within the video frames. We follow DINO (Caron et al., 2021) and evaluate on video frames with a resolution of 480p. We apply label propagation on the attention map from our pretrained model and use mean region-based similarity $\mathcal{J}_m$ and mean contour-based accuracy $\mathcal{F}_m$ as our evaluation metrics.

**GOT-10k: Object tracking**  We evaluate the object-tracking performance of DORA on the GOT-10k dataset (Huang et al., 2019). This is a large-scale benchmark for object tracking that contains 563 categories of common moving objects. The training set contains around 10,000 videos and the test set contains 180 videos. Another challenging aspect of this dataset is that the object classes in the training and test set are non-overlapping. We use the SeqTrack (Chen et al., 2023) codebase to evaluate the performance of different methods on this dataset. In particular, we initialize the encoder weights of SeqTrack with the self-supervised weights and keep them frozen during training. While training, we only update the parameters of the lightweight decoder which consists of 2 transformer blocks. We use all the default hyperparameters. We report mean average overlap (mAO) and success rate (SR) at different thresholds. The mAO measures the class-balanced average overlap between the ground truth and predicted bounding boxes whereas SR indicates the percentage of accurately tracked ground truth bounding boxes where the overlap crosses a certain threshold.

## C  MORE ABLATIONS

**Blurring faces in videos**  We address the privacy concerns that may arise from potential lack of consent during recording by the creator. Given the significant presence of individuals in WTour videos, we employ Deface[‡] to automatically detect and blur faces in WT videos. Specifically, we blur all faces in WT-Amsterdam and observe that DoRA achieves a top-1 accuracy of 45.5% on linear probing, compared to 45.4% without any blur. This demonstrates that face blurring does not impact performance and can effectively mitigate potential privacy and safety issues.

**Pretraining WT video**  We study the effect of pretraining on different videos of WTours. In Table 6a, we observe that the effect is minimal on both image classification and unsupervised object discovery. Notably, the fluctuation in illumination conditions within the *Bangkok* video influences the performance on image classification. It is also interesting to note that, while pretraining on *Amsterdam* is best on image classification, pretraining on *Venice* is best on object discovery. This

---

[‡]https://github.com/ORB-HD/deface

| METHOD | PRETRAIN | EPOCHS | OBJECT DISC. | | SEMANTIC SEG. | | OBJECT DET. | | VOS | |
|---|---|---|---|---|---|---|---|---|---|---|
| | | | JACC | CORLOC | MIOU | $ACC_m$ | MAP | MIOU | $\mathcal{J}_m$ | $\mathcal{F}_m$ |
| DINO (Caron et al., 2021) | IN-1K | 100 | 44.5 | 59.6 | 33.9 | 44.3 | 39.9 | 35.1 | 57.4 | 61.4 |
| DINO (Caron et al., 2021) | IN-1K | 300 | **45.9** | **64.0** | **38.2** | **49.4** | **42.4** | **41.6** | **60.2** | **63.4** |
| DINO (Caron et al., 2021) | $WT_{Venice}$ | 100 | 43.8 | 51.2 | 32.4 | 43.7 | 37.1 | 32.1 | 53.0 | 56.2 |
| DoRA | $WT_{Venice}$ | 100 | 44.0 | 56.2 | 35.4 | 45.5 | 39.5 | 34.7 | 56.4 | 60.4 |
| DoRA | $WT_{Venice}$ | 300 | **44.9** | **56.9** | **37.5** | **47.3** | **42.8** | **42.0** | **57.5** | **60.9** |
| DINO (Caron et al., 2021) | $WT_{all}$ | 100 | 42.9 | 55.8 | 34.1 | 44.0 | 38.3 | 33.2 | 53.3 | 57.6 |
| DoRA | $WT_{all}$ | 100 | 44.3 | 57.1 | 36.9 | 49.0 | 40.7 | 36.3 | 55.1 | 60.2 |
| DoRA | $WT_{all}$ | 300 | **45.2** | **61.3** | **38.7** | **49.8** | **43.7** | **42.6** | **58.4** | **62.0** |

Table 7: *Effect of longer pretraining* ViT-S/16 pretrained, then frozen (object discovery and VOS, same settings as Table 5, Table 5) or fine-tuned (semantic segmentation and object detection, same settings as Table 3). IN-1K: ImageNet-1K.

| # FRAMES | CLASSIFICATION | | OBJECT DISC. | | SEMANTIC SEG. | |
|---|---|---|---|---|---|---|
| | LP | $k$-NN | JACC | CORLOC | MIOU | $ACC_m$ |
| DoRA ($T = 2$) | 43.1 | 32.6 | 41.9 | 53.3 | 32.0 | 43.5 |
| DoRA ($T = 4$) | 44.0 | 32.9 | 42.7 | 54.6 | 32.6 | **44.1** |
| DoRA ($T = 6$) | 44.9 | 33.5 | **44.0** | 55.8 | **32.7** | 43.9 |
| DoRA ($T = 8$) | **45.4** | **33.8** | **44.0** | **56.2** | 32.4 | 43.7 |

Table 8: *Effect of # frames in a video clip.* ViT-S/16 pretrained on $WT_{Venice}$, then frozen features used for classification on ImageNet-1K and object discovery on PASCAL-VOC; features fine-tuned on ADE20K for semantic segmentation.

could be due to the large overlap of objects in these videos with respect to the downstream datasets. However, the consistency of our method across diverse videos indicates that DORA is robust to variations in scenes, number of objects and lighting conditions.

**Presence of cuts**  We now analyse the effect of cuts in representation learning. Cuts are defined as instant transitions from one shot to the next, which is frequent in movies. In action movies, a single shot lasts around 4 seconds, while in romance movies, around 12 seconds on average[§]. To understand the effect of cuts, we compare pretraining on WTours videos and a romance movie. We use PySceneDetect (Castellano) to extract the cut timestamps in the movie and we pretrain DORA by sampling clips that do not intersect cuts; cuts naturally do not exist in WT videos. In Table 6b, we observe that the performance improves significantly in the absence of cuts, as tracking in DORA will fail across a cut.

**Longer pretraining**  We investigate the impact of pretraining DoRA on a single WT video and all WT videos over 300 epochs. The results presented in Table 7 indicate a notable enhancement in performance when DoRA is pretrained for 300 epochs in general. Furthermore, we observe that DoRA pretrained on $WT_{all}$ surpasses $WT_{Venice}$ in VOS pretraining performance, in contrast to the situation where DoRA was pretrained for 100 epochs. It is worth highlighting that the extended pretraining duration also enables DoRA to achieve comparable results to DINO pretrained on ImageNet-1K, while demonstrating superior performance in dense scene understanding tasks.

**Number of frames in a video clip**  We study the effectiveness of DoRA in classification, object discovery and semantic segmentation tasks using a lower number of frames in a video clip, specifically when $T \in \{2, 4, 6\}$. Using the frame-wise loss (8), our goal is to improve the training efficiency of DoRA by achieving comparable performance through pretraining with fewer frames than the default $T = 8$ frames.

We observe from Table 11 that the performance of DoRA in classification and object discovery improves as the number of frames in a video clip increases. Employing $T = 2$ enhances training throughput by 60%, albeit with a 2.3% decrease in linear probing accuracy. However, fine-tuning DoRA for semantic segmentation on ADE20K shows no additional performance gain when varying the number of frames in the video clip. These insights contribute to a detailed understanding of

---

[§]https://stephenfollows.com/many-shots-average-movie/

DoRA's performance across diverse tasks and provide valuable guidance in optimizing its training strategy to suit specific applications.

**Using larger ViT architecture**   We evaluate DoRA (WT-Venice) for 100 epochs on ViT-B/16 for semantic segmentation on ADE20K and object detection on MS-COCO. The results are summarized in the Table below.

| METHOD | ARCH | PRETRAIN | ADE20K | MS-COCO |
|---|---|---|---|---|
| DoRA (ours) | ViT-S/16 | WT-Venice | 35.4 | 39.5 |
| DoRA (ours) | ViT-B/16 | WT-Venice | **40.3** | **41.7** |

Table 9: *Using a larger model.* ViT-B/16 pretrained and then fine-tuned for semantic segmentation on ADE20K and object detection on MS-COCO. We report mIoU on ADE20K and mAP on MS-COCO.

We observe that with ViT-B/16, DoRA achieves 4.9% gain in terms of mIoU on ADE20K and 2.2% in terms of mAP on MS-COCO as compared to using ViT-S/16. This shows that DoRA also scales well with the model size.

## D   MORE COMPARISONS WITH STATE OF THE ART

**Pretraining on ImageNet-1k**   We pretrain DORA on ImageNet-1k and compare with SoTA methods on multiple tasks. Unlike videos, we discover objects but do not track them. Instead, images in a mini-batch are processed independently. Given an input image $\mathbf{X}$, we obtain refined object prototypes as usual (5), but the refined cross-attention (6) is with $\tilde{K}_t$ replaced by $\tilde{K}$ of the same image $\mathbf{X}$. The same image $\mathbf{X}$ is masked for the student (7). The loss is given again by (8) and (9) with $\mathbf{X}_t$ replaced by $\mathbf{X}$, averaged over the mini-batch. We refer to this version as DORA *without tracking* or DORA$^\star$.

DINO (Caron et al., 2021) and iBOT (Zhou et al., 2022a) use only one global crop for the student, while DORA uses $k$ object crops. To compensate, we perform an experiment where we pretrain DORA$^\star$ for 60 epochs and the competitors for 100, thus all methods having the same training time.

From Table 10, we observe that DORA outperforms state-of-the-art self-supervised learning (SSL) methods like DINO and iBOT on image downstream tasks. This demonstrates that the multi-object loss not only enhances performance when pretrained on WTours videos but also achieves superior results when pretrained on ImageNet-1k images.

**Fine-tuning for classification**   We follow the evaluation protocol of iBOT (Zhou et al., 2022a) and fine-tune DINO (ImNet), DoRA (WT-Venice) and DoRA (WT-all) using ViT-S/16 on CIFAR-10/100 (C-10,C-100) (Krizhevsky et al., 2009), iNaturalist18 (iNat 18) (Van Horn et al., 2018), Oxford Flowers (Flwrs) (Nilsback & Zisserman, 2008), Stanford Cars (Cars) (Krause et al., 2013) and ImageNet-1k (ImNet) datasets.

From Table 11, we observe that despite the class distribution of iNat18 being closer to ImNet than WTours, DoRA (WT-Venice) is on-par with DINO (ImNet) when fine-tuned on C-10/C-100, iNat18, Flowers and Cars. Furthermore, DoRA (WT-all) outperforms DINO (ImNet) on all fine-tuning tasks.

| METHOD | EPOCHS | CLASSIFICATION | | OBJECT DISC. | | SEMANTIC SEG. | | OBJECT DET. | |
|---|---|---|---|---|---|---|---|---|---|
| | | LP | $k$-NN | JACC | CORLOC | MIOU | ACC$_m$ | MAP | MIOU |
| DINO (Caron et al., 2021) | 100 | 71.4 | 69.0 | 44.5 | 59.6 | 33.9 | 44.3 | 37.1 | 32.1 |
| iBOT (Zhou et al., 2022a) | 100 | 72.1 | 69.4 | 44.5 | 59.7 | 35.2 | 45.1 | 38.9 | 34.4 |
| DORA$^\star$ (ours) | 60 | 71.9 | 69.4 | 44.4 | 60.0 | 35.4 | 44.9 | 39.3 | 34.9 |
| DORA$^\star$ (ours) | 100 | 72.2 | 69.6 | 44.8 | 60.2 | 35.8 | 45.1 | 39.9 | 35.1 |

Table 10: *Pretraining on ImageNet-1k.* ViT-S/16 pretrained, then frozen (classification and object discovery, same settings as Table 5) or fine-tuned (semantic segmentation and object detection, same settings as Table 3). DORA$^\star$: DORA without tracking; when pretrained for 60 epochs, it has the same training time as DINO and iBOT.

| METHOD | PRETRAINED | C-10 | C-100 | INAT 18 | FLWRS | CARS | IMNET |
|--------|-----------|------|-------|---------|-------|------|-------|
| DINO | IN-1k | 98.7 | 89.8 | 71.5 | 98.3 | 92.2 | 81.3 |
| DoRA (ours) | $WT_{Venice}$ | 98.5 | 89.4 | 69.8 | 94.0 | 92.5 | 80.8 |
| DoRA (ours) | $WT_{all}$ | **98.8** | **89.9** | **72.2** | **98.7** | **93.1** | **81.4** |

Table 11: *Fine-tuning for classification*. ViT-S/16 pretrained, then fine-tuned on different image-based datasets. $WT_{Venice}$: Walking Tours (ours), single video of *Venice*; $WT_{all}$: all videos. IN-1k: ImageNet-1k. We report the top-1 accuracy (%).

We also observe such performance gains when fine-tuning on semantic segmentation on ADE20K and object detection on MS-COCO (Table 3).

## E    MORE VISUALIZATIONS

Figures 5 and 6 show example attention maps obtained using SK on different clips. These figures show that SK (6) leads to attention maps that exhibit spatial locality and are well aligned with objects in the scene. Remarkably, the masks seem to be even robust to occlusions, as shown in the sequence with a bicycle moving behind traffic lights.

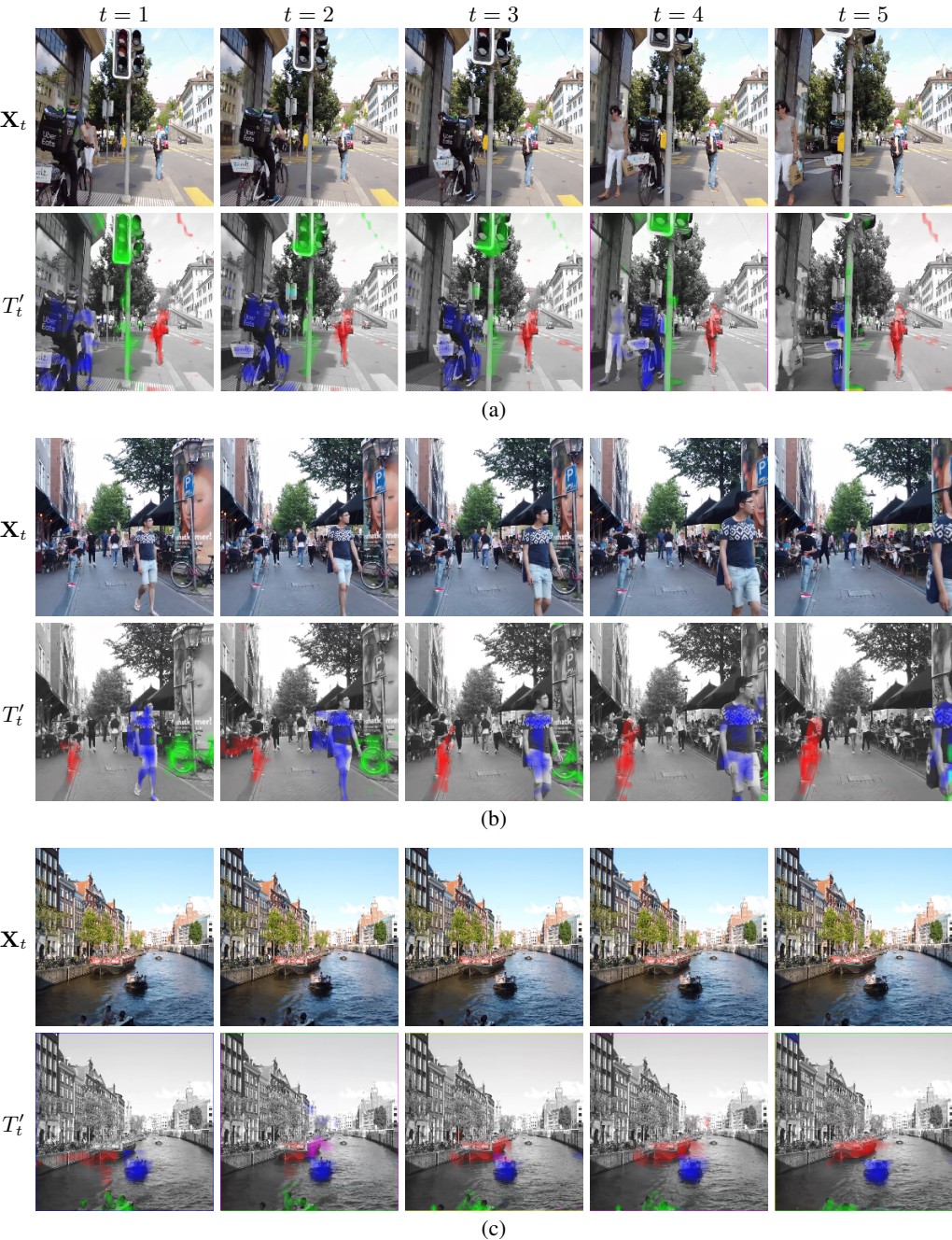

Figure 5: For each input frame $\mathbf{X}_t$ of a video clip, refined cross-attention map $T'_t \in \mathbb{R}^{k \times n}$ (6), using Sinkhorn-Knopp. For each object, one row of $T'_t$ is reshaped as $h/p \times w/p$ and upsampled to an $h \times w$ *attention map* overlaid on the input frame for $k = 3$ objects encoded in blue, red and green channel. $T'_t$ yields three well separated objects shown in blue, red and green.

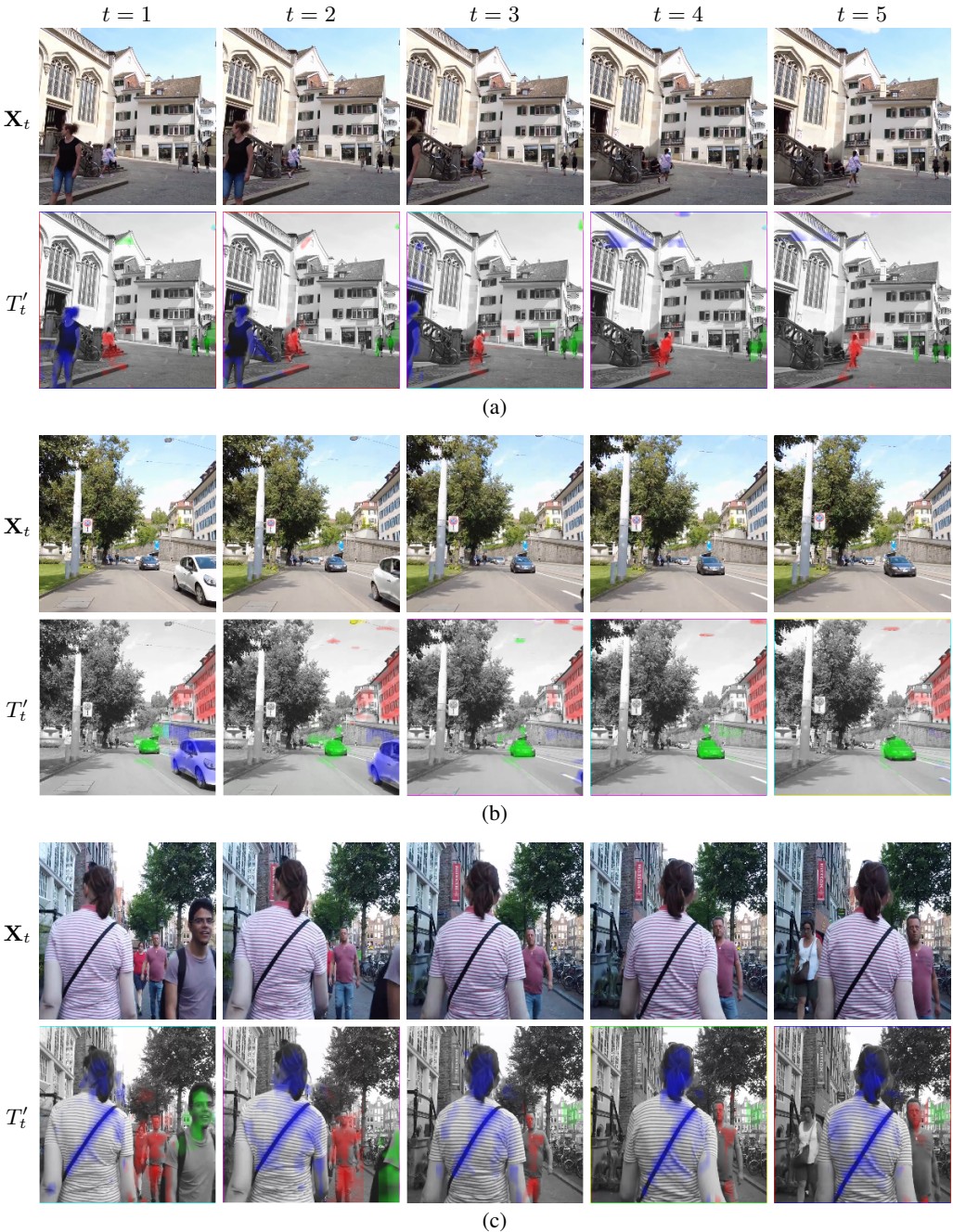

Figure 6: For each input frame $\mathbf{X}_t$ of a video clip, refined cross-attention map $T'_t \in \mathbb{R}^{k \times n}$ (6), using Sinkhorn-Knopp. For each object, one row of $T'_t$ is reshaped as $h/p \times w/p$ and upsampled to an $h \times w$ *attention map* overlaid on the input frame for $k = 3$ objects encoded in blue, red and green channel. $T'_t$ yields three well separated objects shown in blue, red and green.

