# OpenReview forum: "Is ImageNet worth 1 video? Learning strong image encoders from 1 long unlabelled video"
_ICLR.cc/2024/Conference — ICLR 2024 oral_

### Official Review · Reviewer_GQ4C · 2023-10-31

**Soundness:** 2 fair
**Presentation:** 4 excellent
**Contribution:** 3 good
**Rating:** 8
**Confidence:** 5

**Summary:**

This paper investigates self-supervised representation learning from video with a specific focus on the data distributions. In particular, the paper questions the needs of using large-internet scale image datasets and propose instead to learn representation by watching few long videos.

The paper makes two mains contributions:
-	The WTtour datasets which composed by 10 long-videos
-	DORA, a self-supervised representation learning approach that learns to represent and track object in a video at the same time.

The paper evaluates the learned representations on various downstream tasks including ImageNet linear probing, Pascal VOC unsupervised object discovery, MS-COCO/ADE20K object detection/segmentation and DAVIS-2017 for video understanding. DORA pretrained on WTours demonstrates strong performances on ADE20K and MS-COCO

**Strengths:**

- This paper explores the pretraining of visual representation using few long videos which is an original and novel empirical setting.

- They propose a new datasets WTours which could be of interest to the representation learning community.

-  Dora obtains reasonable performance when fine-tuning the performances on ADE-20K and MS-COCO.

**Weaknesses:**

- Performance on ImageNet linear probing seems low. While I understand that video-pretrained models have a disadvantage over image-pretrained model as they can’t be pretrain ‘in-distribution’ with respect to imagenet. However, ImageNet is a standard vision task. It is important to understand why there is such a gap between image and video models on this evaluation.

- The DINO baseline is trained for 100 epochs only which is not the default setting. Additionally, DINO paper reports a performance of 61.8 with a VIT.S/16 on Davis while the paper reports of 54.6 for the same method. I would encourage the authors to report what are the performances of the DINO released models as those models are available.

- DORA shares some similarity with the VITTO. Both approaches learn from video and use an 'unsupervised' pooling mechanism. However, DORA seems to underperform VITTO on the ADE20K and MS-COCO tasks.

- Missing comparison with more recent baselines. It would be nice to add comparison with DINOv2 and a weakly-supervised baseline  OpenCLIP, which are both state-of-art methods.

**Questions:**

I like the motivation and the novel exploration of the paper. However, I think the experimental evaluation could be improved to better support the claims of the paper.

First, I think comparing with state-of-art image-baseline such as DINOv2 and CLIP on the different tasks would really highlight the importance of video pretraining. Second, I think it would be useful to discuss in depth the relation with the VITTO approach. Finally, the current approach currently falls short of image-pretrained model on ImageNet. It would be nice if the author could discuss this limitation in the manuscript.

---

> ### Author Response · Authors · 2023-11-17
> **Response to Reviewer GQ4C**
>
> We appreciate the Reviewer GQ4C's valuable feedback. We address the concerns as follows:
>
> **1. Linear probing performance**
>
> Please refer to *Common response to R-DfLs, R-CACu, R-GQ4C: Results on Imagenet linear probing*.
>
> **2. DAVIS numbers**
>
> DINO (Caron et al.) reports a performance of 61.8 on DAVIS using ViT-S/16 when pretrained on *ImNet* for 300 epochs. In our work, we report that DINO achieves a performance of 54.6 on DAVIS when pretrained on *WT-Venice* for 100 epochs. The difference in performance is from the different pretraining dataset and higher number of pretraining epochs.
>
> In Section C under ``Longer pretraining" and Table 7 in the Appendix, we have added the comparison of DoRA vs DINO when pretrained for 300 epochs on WT-Venice and WT-all.
>
> **3. DoRA vs VITO**
>
> VITO is pretrained for 300 epochs using Resnet-50 on VideoNet: a curated dataset of 1 million videos whose distribution is similar to that of ImNet. While DoRA is pretrained for 100 epochs using ViT-S/16 on 1 or 10 long uncurated WTour videos whose distribution is different from ImNet.
>
> **4. Comparison with Dino-V2 and OpenCLIP**
>
> Thanks for this suggestion. We will add experiments with DINO-v2 and OpenCLIP in the camera ready version

---

> > ### Comment · Area_Chair_rbJh · 2023-11-19
> >
> > Dear Reviewer,
> >
> > The author has provided responses to your questions and concerns. Could you please read their responses and ask any follow-up questions, if any?
> >
> > Thank you!

---

> > ### Comment · Reviewer_GQ4C · 2023-11-20
> >
> > Thank you for your answers. The rebuttal did address most of my concern and I will update the paper score accordingly.
> >
> > I would encourage the author to add DINO-v2, OpenCLIP baselines for camera ready.
> >
> > Thanks!

---

> > > ### Author Response · Authors · 2023-11-21
> > > **Thank you R-GQ4C**
> > >
> > > We sincerely appreciate R-GQ4C's time and effort in evaluating our work, for the insightful comments, and for raising the score to "8". We shall add the additional experiments on DINO-v2 and OpenCLIP to the camera-ready version of the paper.

---

### Official Review · Reviewer_krCr · 2023-11-01

**Soundness:** 3 good
**Presentation:** 3 good
**Contribution:** 4 excellent
**Rating:** 8
**Confidence:** 4

**Summary:**

This work introduces a method for self-supervised learning of image representation models. It is based on a similar scheme to DINO (Caron et. al. 2021) that learns image representation by distilling a moving average teacher model's representation of the global image view to the representation of a student model's on multiple local views. The novel way of achieving this in this work is to use the feature correspondence provided in hours-long walking tour videos to provide tracking, which can identify the location of different objects in any video frame without annotation. This object-centric way of generating local views seems to lead to good learned representation, which is examined in the experimental section.

In total, the proposed method does not need curated image or video data for self-supervised learning and can learn effective representation from hours-long walking tour videos. The proposed multi-object masking approach is shown to be significant in learning effective visual representation.

**Strengths:**

+ The fact that this method does not use curated data is a big plus for me. The way of collecting this type of data seems to be easily scalable, as it can be from walking tours or vehicle-mounted cameras.

+ Using correspondence-based tracking to provide localization is sound and practical in representation learning.

+ The experimental results show the model trained with the proposed method on the walking tour videos can outperform strong baselines trained on curated datasets such as ImageNet and Kinetics-400.

**Weaknesses:**

- One minor issue I have about the presentation is the introduction of the tracking module. The tracker is not learned, and it is only used to provide object locations. I would like further discussion on the potential use of the corresponding information. Also, a comparison on the effect of using different types of unsupervised trackers would also help strengthen the work as the major idea seems to be not dependent on a certain type of tracker.

**Questions:**

I have the following questions after reading the text:

1. In Eq. (8) the learning is done on a single frame. Because the tracking has already provided corresponding between locations across multiple frames, is there a certain consideration to not use views from multiple frames in this loss function?

2. The authors have presented 10 walking tour videos. The results in Table 5 suggest training on one video already achieves similar accuracy obtained by training on all videos. Does this mean one video is sufficient? Is there some point in further scaling up the training data? I would like to see a discussion on this topic.

---

> ### Author Response · Authors · 2023-11-17
> **Response to Reviewer krCr**
>
> We appreciate the Reviewer krCr's valuable feedback. We address the concerns as follows:
>
> **1. Using unsupervised trackers**.
>
> Thanks for this interesting suggestion. We shall integrate UnSupTrack [1*] with DoRA. We shall first detect the objects using our proposed multi-object masking method, which we then use as input to UnSupTrack. We shall add this result in the camera-ready version.
>
> [1*] Karthik *et al.*, Simple Unsupervised Multi-Object Tracking, ECCV 2020
>
> **2. Views from multiple frames**.
>
> Thanks for the interesting question. We will apply the loss function between the global crop of the teacher network (from the reference frame $t_0$ that is used in tracking) to multi-object crops of the student network from all other frames $t$ in the mini-batch. Due to time and compute constraints, we will add this experiment to the camera-ready version.
>
> **3. WT-all vs. WT-1vid**
>
> Please refer to *Common response to R-DfLs, R-CACu, R-krCr: WT-all vs. WT-venice (1 video)*

---

> > ### Comment · Area_Chair_rbJh · 2023-11-19
> >
> > Dear Reviewer,
> >
> > The author has provided responses to your questions and concerns. Could you please read their responses and ask any follow-up questions, if any?
> >
> > Thank you!

---

> > > ### Comment · Reviewer_krCr · 2023-11-23
> > >
> > > I thank the authors for their responses to my questions. My question regarding the data scalability is addressed.  I want to maintain my initial rating.
> > >
> > > I would be curious to see the results of the multi-frame experiments in the revised version.

---

### Official Review · Reviewer_CACu · 2023-11-01

**Soundness:** 4 excellent
**Presentation:** 3 good
**Contribution:** 3 good
**Rating:** 8
**Confidence:** 4

**Summary:**

This paper proposed a new perspective on self-supervised learning (SSL). Instead of pretraining models on ImageNet-like object-centric datasets, the paper pretrained the models on egocentric videos (“Walking Tours” dataset) which depict numbers of objects and are comparable with human learning. Compared with other video datasets for SSL, Walking Tour dataset had more objects and classes and more gradual shifts in lightness. To pretrain on Walking Tours, the paper proposed a novel SSL method, based on DINO, to first discover objects and then track objects, named DoRA. In every batch, DoRA randomly sampled 8 frames temporally separated by 1 second, discovered objects in the first frame, and tracked them over the following 7 frames. In the default setting, objects are tracked by cross-attention in the multi-object tracker, which leads to spatially overlapping. The paper then proposed to establish object-patch correspondences using the Sinkhorn-Knopp algorithm to deal with the problem. After finding separated objects, the input video clip in the student branch would be masked to contrast with the clip in the teacher branch. Experiments on dense prediction tasks show that DoRA on Walking tours shows comparable performance with other SSL methods pretrained on ImageNet.

**Strengths:**

1.	This paper proposed a new pretraining method on egocentric videos which are uncurated and comparable with human learning. As these videos do not involve human annotation, they can be easily obtained, leading to a promising way of SSL.
2.	The proposed method is simple and neat. DoRA provides an intuitive but effective way to learn from frames that contain multiple objects.

**Weaknesses:**

1.	The paper mainly talks about how we can learn discriminative representations from egocentric videos, whereas what kind of egocentric videos are suitable for DoRA is not deeply discussed. It would contribute more to the community if we knew what properties a video should have to be worth learning.
2.	A good SSL method should be scalable, not only on the dataset but also on the model structure. It would be better for authors to show more results on larger ViTs.
3.	Some minor writing problems. (1) in Sec. 4 “Discovering objects with multi-head attention”, $\widetilde{Q}$, $\widetilde{K_t}$ are only defined in Fig.3 and are not defined in text. (2) In Fig.3 (Left), the input should be $X_t^{o_i}$

**Questions:**

1. In Table 4, why does DoRA perform worse when using WT_all than when using WT_Venice?
2. DoRA shows inferior performance on ImageNet linear probe (LP) but superior performance on dense prediction tasks, would it perform better than other contrastive methods on the ImageNet fine-tuning task? Like MAE [1], lower on LP but higher on fine-tuning.

[1] He, Kaiming, et al. "Masked autoencoders are scalable vision learners." Proceedings of the IEEE/CVF conference on computer vision and pattern recognition. 2022.

**Details Of Ethics Concerns:**

None.

---

> ### Author Response · Authors · 2023-11-17
> **Response to Reviewer CACu**
>
> We appreciate the Reviewer CACu's valuable feedback. We address the concerns as follows:
>
> **1. Type of egocentric videos**.
>
> In Table 2(a), we show that the performance of DoRA is not specific to egocentric videos, *e.g.*, WTours and Epic-Kitchens (EK), but also achieves good performance on diverse pretraining datasets like Kinetics-400 (K-400) and Movie videos (Movie$_{\text{rom}}$). Thus, we observe that DoRA is agnostic to type of video pretraining dataset. Our argument for videos like WTours is that they can collected or filmed very easily.
>
> **2. Using larger ViT**.
>
> Please refer to *Common response to R-DfLs, R-CACu: Using larger ViT*
>
> **3. WT-all vs. WT-Venice**
>
> Please refer to the response in *Common response to R-DfLs, R-CACu, R-krCr: WT-all vs. WT-venice (1 video)*.
>
> **4. Linear Probing**
>
> Please refer to the response in *Common response to R-DfLs, R-CACu, R-GQ4C: Results on Imagenet linear probing*.
>
> **5. Minor corrections in Figure 3**
>
> Thanks for pointing this out. We have corrected it in the revised version.
> We define $\tilde{Q} \in \mathbb{R}^{n \times d}$ in the sentence above eq(2) and $\tilde{K} \in \mathbb{R}^{n \times d}$ in the sentence below eq (3)

---

### Official Review · Reviewer_DfLs · 2023-11-01

**Soundness:** 3 good
**Presentation:** 3 good
**Contribution:** 2 fair
**Rating:** 6
**Confidence:** 4

**Summary:**

The paper proposed a dataset *Walking Tours (WT)* consisting of high definition street-walk videos of 23 hours total length, and DoRA, a multi-object-tracking-inspired framework to learn visual representation from different views of the same objects in adjacent frames. The proposed method largely follows the DINO framework, with the local crops in DINO replaced with a tracking of objects in a video. The method is tested on several mainstream visual tasks, including image detection and segmentation, video object segmentation, object tracking, image classification and object discovery. Compared to its baseline DINO, the proposed method outperforms it clearly when using the same WT dataset.

**Strengths:**

* The motivation to learn visual representation using of the appearance variation of object along time is sound and is probably worth exploring in the future as an additional source of information in self-supervised visual learning.

* The proposed architecture to utilize temporal information by tracking the same object in different frames is novel, and the visualization can justify that the proposed method is working as expected.

**Weaknesses:**

* **Scalability concern.** Although the proposed method does outperform DINO on all reported experiments in controlled experiments (i.e., with the proposed WT dataset), the results with even WT_{all} is still not clearly better than DINO with ImageNet-1k. This leaves it unknown whether the WT dataset will eventually outperform ImageNet-1k (or the even larger ones like ImageNet-22k and LVD-142M) with the dataset at a reasonable scale. Also the experiments are mostly on ViT-S, which is relatively small compared to the well-known works in the field (which usually report at least ViT-B), so it is also hard to tell whether the proposed method scale well with the model size.

* **Significantly worse results on image classification.** I have noticed that the image classification results of WT-pretrained models are lower than ImageNet-1k-pretrained by a fairly large margin (45LP / 36KNN on WT vs. 72LP / 70KNN on ImageNet). Although one can argue that this is because of the domain gap between WT and IN, I would consider the accuracy difference large enough to require some formal justification (e.g., run WT and IN pretrained models on a 3rd classification dataset like iNaturalist or Places) to conclude that the WT-pretrained models are not significantly weaker on image classification tasks.

* **The potential privacy and safety issues of the dataset.** Also see *Details Of Ethics Concerns*. As the paper claims the dataset as a main contribution, I would expect more efforts in assessing the privacy and safety issues in the dataset (e.g., How many clear faces are detected and what are their resolutions? How many harmful scenes are detected to the best effort of the authors?) and clarifying the legal issues and usage restrictions of the dataset (e.g., Is it possible that some videos are taken down upon the request from people appearing in them? Is their usage in some jurisdictions not allowed / limited to non-commercial only? What are some possible negative effects if the models remember the private information in it? What are the possible effects of some common mitigations, like blurring the faces in Google street view?)

**Questions:**

* In addition to training epochs, it would be kind to also mention the actual training time as a more practical measurement of the training cost.

* In Appendix D, are there any other differences between DoRA without tracking and DINO other than the crop generation method?

**Details Of Ethics Concerns:**

The paper proposes a new dataset consisting of 23 hours of UHD (4k) videos filmed on the public streets which may contain personal information like high resolution faces of strangers and audio recordings of the nearby people talking (very likely) without their consent. Although the videos are not filmed by the authors themselves and are in CC-BY licenses on YouTube according to the paper, I'm concerned that it needs a careful discussion regarding the compliance issues or restrictions of using them for machine learning purposes (or even posting them on YouTube in the first place) in different jurisdictions.

There is also no assessment or mitigation about the potentially harmful scenes (e.g., violent, harassing, criminal) in the proposed dataset.

---

> ### Author Response · Authors · 2023-11-17
> **Response to Reviewer DfLs**
>
> We appreciate the Reviewer DfLs's valuable feedback. We address the concerns as follows:
>
> **1. Stability**
>
> Please refer to the response in *Common response to R-DfLs, R-CACu, R-krCr: WT-all vs. WT-venice (1 video)*.
>
> **2. Larger ViT.**
>
> Please refer to the response in *Common response to R-DfLs, R-CACu: Using larger ViT*.
>
> **3. Linear probing**
>
> Please refer to the response in *Common response to R-DfLs, R-CACu, R-GQ4C: Results on Imagenet linear probing*.
>
> **4. Potential privacy and safety issues of the dataset**
>
> The reviewer makes a very good point. To address this concern, we use Deface (https://github.com/ORB-HD/deface) to automatically detect and blur faces in WT videos. Using these modified WT videos, we apply DoRA on WT-Venice. We shall report the results once pretraining is completed.
>
> **5. Training Cost**
>
> Please refer to the response in *Common response to R-YYAs, R-DfLs: Training costs*
>
> **6. Difference between DoRA w/o tracking and DoRA with tracking.**
>
> In Appendix D, we apply equation (6) on a single image rather than a set of frames, i.e., there is no tracking involved when DoRA is pretrained on ImNet. Thus, other than the crop generation method, there is no other difference between DoRA* (DoRA without tracking) and DINO.

---

> > ### Comment · Area_Chair_rbJh · 2023-11-19
> >
> > Dear Reviewer,
> >
> > The author has provided responses to your questions and concerns. Could you please read their responses and ask any follow-up questions, if any?
> >
> > Thank you!

---

> ### Comment · Reviewer_DfLs · 2023-11-23
>
> Thanks for the responses from the authors and the comments from the other reviewers.
>
> My concerns regarding the technical issues are mostly addressed: Despite that not all experiments are finished in the short period of the rebuttal, I do believe the method looks more promising with the current information and would raise my rating based on the improvements.
>
> I also appreciate very much the authors' efforts to resolve the ethical issues. Given that it has received emphasis only in the recent one or two years, I actually do not expect some specific experiments, but consider it good enough to give some general discussions about the best practices or the potential risks as a reminder for the future users of the dataset.
>
> Considering the factors above I have raised the score to a weak accept.

---

### Official Review · Reviewer_YYAs · 2023-11-03

**Soundness:** 3 good
**Presentation:** 3 good
**Contribution:** 3 good
**Rating:** 8
**Confidence:** 4

**Summary:**

The authors consider how both (a) data and (b) method can improve training an image encoder in a self-supervised manner. Regarding data, they introduce an open-source dataset containing long, first-person videos, and propose several advantages of this over curated image (and video) datasets. Given this dataset contribution, the authors propose a self-supervised method which tracks objects to act as a signal for a classical multi-view self-supervised learning (SSL) loss. This method leverages some key properties from the dataset, specifically that the videos have natural scene transitions and are of a person-person view. Instead of using off-the-shelf object-trackers or optical flow to establish correspondence, the authors use the attention-map between the [CLS] token from a selection of heads and the patch-embeddings. Optimal transport is used to establish unique object-patch correspondence (i.e. non overlapping patches) and then given this multi-view correspondence, a technique based on DINO is used.

Overall the authors show that for many downstream tasks, their method (DORA) pre-trained on (even one) video where the number of frames is comparable to imagenet-1K (IN-1K) achieves better performance than DINO pretrained on IN-1k. In comparison when DINO is pretrained on the same data, the peformance is worse than IN-1K which suggests it is the unique coupling of dataset and training-method which helps the authors achieve SOTA results

**Strengths:**

- The paper is well written and the supplementary work provides a good level of detail and convincing ablations and visualisations
- The contribution of the open-source dataset (10 videos) to replicate this work is very useful for the community
- The statistical analysis of the contributed dataset also useful
- This is a nice use-case of Sinkhorn–Knopp to avoid having to use non end-to-end approaches like optical-flow or off-the-shelf object-detectors and the motivation which shows overlapping spatial regions when linearly projecting the attention-map (instead) is a good argument for its use
- Overall, the results with DORA are very impressive

**Weaknesses:**

- I'm a bit confused about Table 4: Video Object Segmentation (DAVIS-2017). In the DINO paper they report ViT-S/16 with INet getting 61.8, 60.2, 63.4 respectively (their Table 5), however your Table 4 reports DINO as getting 59.4, 57.4, 61.4 what accounts for this difference?

**Questions:**

- What is the stability like when using SK, aside from the entropy regularisation is some annealing schedule needed that transforms the coupling matrix from soft to hard gradually?
- Is there any intuition why the k-attention maps obtained by projecting the attention map are spatially overlapping? Is there any possibility to use a simple heuristic to avoid it that can be ablated with SK?
- What is the training cost of using SK for every forward-pass like this?

---

> ### Author Response · Authors · 2023-11-17
> **Response to Reviewer YYAs**
>
> We appreciate the Reviewer YYAs's valuable feedback. We address the concerns as follows:
>
> **1. Difference in VOS numbers.**
>
> In Table 5 (Caron et al.), the authors evaluate DINO (ImNet) for *300 epochs* for Video Object Segmentation on DAVIS 2017. In Table 4 of DoRA, we reproduce DINO (ImNet) for *100 epochs*. Due to the difference in training epochs, we observe a difference in performance 59.4 (100 epochs) vs. 61.8 (300 epochs).
>
> **2. Stability of SK**
>
> SK is an iterative optimization algorithm that indeed converges, with its cost function monotonically decreasing with the number of iterations.
>
> We understand that the reviewer asks about training stability when using SK---if not, please clarify. We empirically find that, when using SK with the features from the last layer of the transformer to compute refined object prototypes $P'$, training is indeed stable.
>
> Using annealing to transform the coupling matrix from soft to hard might be needed if we target hard assignment between object prototypes and patch features. Hard assignment can also be achieved by using a smaller value of $\epsilon$ (coefficient of entropy regularizer), which improves one-to-one matching, although it makes optimization harder. We evaluate the performance of DoRA with a smaller value of $\epsilon$ and increasing the number of iterations to 60 (default iterations is 30), observing sub-optimal results on downstream tasks. This indicates that downstream task performance does not benefit from hard assignment.
>
> **3. Overlapping attention maps.**
>
> The overlap of the attention maps obtained from different heads is commonly observed as there is no supervisory signal during training to ensure diverse attention from different heads. For example, see Fig. 10 of DINO (Appendix), where the authors show overlap for the different heads in the last layer.
>
> In our previous experiments, we used a simple heuristic inspired from CutLer [2*]. This is an unsupervised object discovery method that iteratively uses normalized cuts on patch affinity matrix to find foreground objects. Similarly, we iteratively removed attended regions to find more, non-overlapping ones. However, on WT videos, we observed that this heuristic did not achieve consistent improvements on all downstream tasks. In particular, its results were sub-optimal on linear probing on Imagenet and unsupervised object discovery on Pascal VOC.
>
> [2*] Wang *et al.*, Cut and Learn for Unsupervised Object Detection and Instance Segmentation, CVPR 2023.
>
> **4. Training Cost**
>
> Please refer to the response in *Common response to R-YYAs, R-DfLs: Training costs*

---

> > ### Comment · Area_Chair_rbJh · 2023-11-19
> >
> > Dear Reviewer,
> >
> > The author has provided responses to your questions and concerns. Could you please read their responses and ask any follow-up questions, if any?
> >
> > Thank you!

---

> > > ### Comment · Reviewer_YYAs · 2023-11-21
> > >
> > > Dear authors, thank you for the clarification regarding your reproduction of DINO and further comments about the stability of the SK algo. I have increased my rating from weak accept to accept.

---

> > > > ### Author Response · Authors · 2023-11-21
> > > > **Thank you R-YYAs**
> > > >
> > > > Thank you very much. We sincerely appreciate your time and effort in evaluating our work, for the insightful comments, and for raising the score to "8".

---

### Author Response · Authors · 2023-11-17
**Common response to R-YYAs, R-DfLs: Training costs**

We thank the reviewers for this interesting comment. As per their suggestion, we compute the training throughput and the overall training training time of DoRA (WT-Venice) and DINO (WT-Venice). On the NVIDIA A100-80GB GPU, DINO achieves an training throughput of 1907 im/sec and DoRA achieves 1598 im/sec, averaged over 10 runs. For a single epoch, DINO uses 4hr and 5min of pretraining time on 8 A100-80GB GPUs, while DoRA uses 6 hr and 48 min.

To compensate for the increase in training time, we compare DoRA pretrained for 60 epochs with DINO pretrained for 100 epochs on ImNet in Appendix D and Table 8. We show that the improvement in performance is due to our multi-object masking.

The computational overhead due to SK is minimal as DoRA uses only 30 SK iterations. The throughput of DoRA (without SK) is 1883 im/sec, while DoRA (with SK) is 1860 im/sec.

---

### Author Response · Authors · 2023-11-17
**Common response to R-DfLs, R-CACu, R-GQ4C: Results on Imagenet linear probing**

As R-DfLs and R-GQ4C point out, because of the domain gap between WT dataset and ImageNet, there is a large gap for linear probing on ImageNet itself. As per the reviewers suggestion, we follow the evaluation protocol in iBOT and finetune DINO (ImNet), DoRA (WT-Venice) and DoRA (WT-all) using ViT-S/16 on CIFAR-10/100 (C-10,C-100), iNaturalist18 (iNat 18), Oxford Flowers (Flwrs), Stanford Cars (Cars) and ImageNet-1k (ImNet) dataset. The results are given in the table below:

| Method         | Pretrained   | C-10  | C-100 | iNat 18 | Flwrs | Cars | ImNet |
| -------------- | ------------ | ----- | ----- | ------- | ----- | ---- | ----- |
| DINO           | ImNet        | 98.7  | 89.8  | 71.5    | 98.3  | 92.2 | 81.3  |
| DoRA (ours)    | WT-Venice    | 98.5  | 89.4  | 69.8    | 94.0  | 92.5 | 80.8  |
| DoRA (ours)    | WT-all       | **98.8** | **89.9** | **72.2** | **98.7** | **93.1** | **81.4** |



Despite the class distribution of iNat18 being closer to ImNet than WTours, we observe that DoRA (WT-Venice) is on-par with DINO (ImNet) when fine-tuned on C-10/C-100, iNat18, Flowers and Cars datasets. Furthermore, DoRA (WT-all) outperforms DINO on ImNet on all fine-tuning tasks. We also observe such performance gains when finetuned on semantic segmentation on ADE20K and object detection on MS-COCO (Table 3 of main paper), similar to MAE.


Additionally, we also perform scene classification on Places205, measuring classification accuracy, using linear probing on models pre-trained on WT-Venice and WT-all for 100 epochs. The results are in the table below.

| Method         | Pretrained   | top-1 \% |
| -------------- | ------------ | -------- |
| DINO           | ImNet        | **54.5**  |
| DoRA (ours)    | WT-Venice    | 49.3     |
| DoRA (ours)    | WT-all       | 51.8     |


These results indicate that the gap on linear probing on Imagenet is due to the domain gap and not a gap between images and video. We thank the reviewers for this very interesting comment. We shall add these results.

---

### Author Response · Authors · 2023-11-17
**Common response to R-DfLs, R-CACu, R-krCr: WT-all vs. WT-venice (1 video)**

Indeed, DoRA (WT-all) is on-par with DoRA (WT-Venice) on linear probing on ImNet. However, in terms of $k$-NN on ImNet, DoRA (WT-all) outperforms DoRA (WT-Venice) by 2\%. In ``Common response to R-DfLs, R-CACu, R-GQ4C: Results on ImageNet linear probing", we show that DoRA (WT-all) outperforms DoRA (WT-Venice) and DINO (ImNet) when fine-tuned on 5 classification tasks. We also observe in Table 3 that DoRA gives superior performance when fully fine-tuned on spatially dense tasks, similar to MAE.

In Table 4 (object tracking), DoRA (WT-all) outperforms DoRA  (WT-Venice) by 4.5\% in mAO,
and 5.5\% in SR$_{0.75}$. Additionally, in Table 3 (semantic segmentation on ADE20K), we observe that DoRA (WT-all) outperforms DoRA (WT-Venice) by 1.5\% in mIoU and 2.5\% in terms of Acc$_m$. In the same table (instance segmentation on MS-COCO), DoRA (WT-all) is better by 1.6\%.


The drop in performance of DoRA (WT-all) with respect to DoRA (WT-Venice) in Table 4 (video object segmentation) refers only to a single downstream task out of seven in the paper.


Finally, we have added more results in Section C under ``Longer pretraining" and Table 7 in the Appendix. These new results show that, when pretrained for 300 epochs (instead of 100 epochs), DoRA (WT-all) significantly outperforms DoRA (WT-Venice) on image based downstream tasks.

---

### Author Response · Authors · 2023-11-17
**Common response to R-DfLs, R-CACu: Using larger ViT**

We evaluate DoRA (WT-Venice) for 100 epochs on ViT-B/16 for semantic segmentation on ADE20K and object detection on MS-COCO. The results are summarized in the Table below.

| Method         | Arch       | Pretrain    | ADE20K | MS-COCO |
| -------------- | ---------- | ----------- | ------ | ------- |
| DoRA (ours)    | ViT-S/16    | WT-Venice   | 35.4   | 39.5    |
| DoRA (ours)    | ViT-B/16    | WT-Venice   | **40.3** | **41.7** |


We observe that with ViT-B/16, DoRA achieves 4.9\% gain in terms of mIoU on ADE20K and 2.2\% in terms of mAP on MS-COCO as compared to using ViT-S/16. This shows that DoRA also scales well with the model size. We thank the reviewers for this suggestion. We shall add the results with ViT-B/16 for all downstream tasks.

---

### Meta-Review · Area_Chair_rbJh · 2023-11-27

**Metareview:**

The paper presents a self-supervised setting on egocentric videos and introduces an egocentric video dataset to demonstrate its use cases. Moreover, the paper presents a novel method to learn from continuous videos based on "tracking to learn to recognize" approach.

After taking into account the five reviews and discussions among reviewers, the strengths and weaknesses are summarized below:

Strengths summarized below:
1. the paper is well-written and organized; the problem is well-motivated and less studied in the literature
2. the data and source code will be made available, useful for reproducibility
3. contribution of a new egocentric video dataset with distinct features from other existing datasets
4. contribution of a novel method leveraging on "tracking to learn to recognize".
5. results in several down-stream tasks are impressive


Weakness summarized below:
1. ethical concerns due to the new video dataset contributions
2. slightly worse results on image classification, such as on imagenet datasets, ADE20K, and MSCOCO tasks; justification about lower performance is needed
3. more explorations are needed to test what type of egocentric videos are suitable for DoRA (the proposed model) or general self-supervised learning methods on videos

**Justification For Why Not Higher Score:**

N/A

**Justification For Why Not Lower Score:**

Contributions significantly outweigh the weaknesses or limitations.
The contributions are impactful to the broad community of computer vision, machine learning, cognitive science, and artificial intelligence.

---

### Decision · Program_Chairs · 2024-01-16

Accept (oral)